environmental science

Bonn Challenge, New York Declaration on Forests, degradation, deforestation, ecosystem functioning, sustainable development goals

**Author for correspondence:**
John A. Stanturf
e-mail: johnalvin.stanturf@emu.ee, drdirt48@gmail.com

Contribution to Special Collection on Sustainable Land Use.

# Forest landscape restoration: state of play

John A. Stanturf[1,2] and Stephanie Mansourian[3,4]

[1]Institute of Forestry and Rural Engineering, Estonian University of Life Sciences, Kreutzwaldi 5, 51014 Tartu, Estonia
[2]InNovaSilva, Højen Tang 80, 7100 Vejle, Denmark
[3]Mansourian.org, 36 Mont d'Eau du Milieu, 1276 Gingins, Switzerland
[4]University of Geneva, Geneva, Switzerland

JAS, 0000-0002-6828-9459; SM, 0000-0002-0897-514X

Tree planting has been widely touted as an inexpensive way to meet multiple international environmental goals for mitigating climate change, reversing landscape degradation and restoring biodiversity restoration. The Bonn Challenge and New York Declaration on Forests, motivated by widespread deforestation and forest degradation, call for restoring 350 million ha by 2030 by relying on forest landscape restoration (FLR) processes. Because the 173 million ha commitments made by 63 nations, regions and companies are not legally binding, expectations of what FLR means lacks consensus. The frequent disconnect between top-level aspirations and on-the-ground implementation results in limited data on FLR activities. Additionally, some countries have made landscape-scale restoration outside of the Bonn Challenge. We compared and contrasted the theory and practice of FLR and compiled information from databases of projects and initiatives and case studies. We present the main FLR initiatives happening across regional groups; in many regions, the potential need/opportunity for forest restoration exceeds the FLR activities underway. Multiple objectives can be met by manipulating vegetation (increasing structural complexity, changing species composition and restoring natural disturbances). Livelihood interventions are context-specific but include collecting or raising non-timber forest products, employment and community forests; other interventions address tenure and governance.

# 1. Introduction

Plant a tree and save the world! This is an over-simplified version of the widely expressed goals of planting a trillion trees or trees on a billion hectares (ha). Tree planting has motivational appeal and great potential to meet multiple international environmental goals [1], particularly international treaties addressing climate mitigation (UN Framework Convention on Climate Change), landscape degradation (UN Convention to Combat Desertification) and biodiversity conservation (UN Convention on Biological

Diversity), as well as a role in watershed protection (UN Office for Disaster Risk Reduction) [2–6]. In addition, trees and forests are critical to sustaining many human communities, as recognized in the UN Sustainable Development Goals [7,8]. These varied goals culminated in the 2021–2030 UN Decade of Ecosystem Restoration [9].

Forests cover 31% of the world's land surface, just over 4 billion ha; human activity has reduced this from a pre-industrial estimate of 5.9 billion ha of forest cover [10]. The importance of forests and continued threats to them from multiple natural and anthropogenic drivers was met in 2011 with the high-profile policy response of the Bonn Challenge [11] to restore 150 million ha of forests by 2020, that was expanded to 350 million ha by 2030 in 2014 by the New York Declaration on Forests [12]. The Bonn Challenge (BC) has spawned regional offshoots in Africa [13], Latin America and the Caribbean [14], and the Caucasus and Central Asia [15], and discussions are underway in other regions. All of these efforts are underpinned by the forest landscape restoration (FLR) approach, a process that seeks to regain ecological integrity while enhancing human well-being [16]. Although restoring forests is clearly needed, tree planting is more complex than often understood [17,18], and should not be an excuse for continued forest loss and degradation or for the afforestation of non-forested ecosystems [19,20]. Ambitious targets risk turning FLR into an end in and of itself, rather than one of many tools to ensure that our planet maintains its forests.

Here, we provide a brief overview of the historical development of FLR, introduce the FLR 'ideal project' and the projected potential of FLR to realize the aspirational goals for the BC. Data limitations preclude a comprehensive view of the reality of what is happening globally; nevertheless, we attempt a description of the scope of current and projected FLR activity by compiling information from regional overviews and lessons learned from country case studies. Our intention is to better understand what is happening in diverse regions under the banner of FLR, rather than an assessment of the quality of these interventions (often impossible to do correctly for the lack of effective monitoring and detailed reporting).

The BC is not the whole story of FLR; significant FLR-type efforts are underway or have been conducted in many countries that have not committed to the BC. There are multiple entry points for FLR: restoring ecological functioning, improving biodiversity, or job creation and livelihoods, etc. Yet, FLR by definition has multiple social and ecological objectives. At the same time, there are numerous categories of interventions that return tree cover, which may not qualify as FLR (see [19]); not every restoration intervention requires the label FLR. In fact, in many cases, actions that do not meet the FLR principles [21] are being erroneously labelled as FLR [22]. To illustrate the potential scope of FLR, we include significant landscape-scale efforts being made in some countries that have not made formal BC commitments.

Significant obstacles to realizing the potential of FLR include superficial knowledge of the problem and solutions. The concepts of degradation, restoration and FLR lack consensus understanding outside of policy and/or academic circles [23–27], and many national languages lack words for forest restoration. The importance of local context adds complexity and nuance that can be overlooked in national and international, top-down restoration programmes. Overcoming these limitations begins with recognizing three important details that must be at the forefront for FLR to be effective [1] degraded areas are not necessarily uninhabited; they are degraded because people are using them and unless local needs are addressed, FLR is likely to fail [2,28] increasing forest cover/area is not the whole story; in many countries, forests remain but are degraded by over-exploitation, invasive species, altered fire regimes, and other disturbances [29–32] and are important targets for restoration; and [3] most importantly, the drivers of deforestation and degradation, both direct and indirect [33–40], must be addressed. Increasing forest cover in areas where forests do not belong is also not compatible with the FLR approach [21,41,42]. FLR requires planning at large scales (landscapes) to accommodate multiple objectives, but specific restoration interventions take place at the local level and, to be sustainable, must meet the needs of, and benefit local communities/stakeholders [28]. Above all, restoring a landscape requires time [43,44].

## 2. Methods

The purpose of our research was to compare the theory and practice of FLR as illustrated by national initiatives under the Bonn Challenge (and related) commitments. We reviewed founding and historical documents of the FLR movement, tracing it back to the definition of the concept [16]. Data were also collected through our participation in various FLR-related events starting with the first workshop that

defined the term in 2000 through to more recent meetings such as during the World Conservation Congress in 2016, the Global Landscape Forum in 2017, the IUFRO World Congress in Curitiba in 2019 and the Society for Ecological Restoration's International Conference that same year. We reviewed online databases of projects and initiatives as well as the Bonn Challenge website, those of the AFR100 and the Initiative 20×20. Our research was exploratory and not systematic as the intention was to obtain an overview rather than a complete registry of projects. Our analysis focused, on the one hand, on identifying restoration potential and, on the other hand, contrasting theory and practice. First, we provide a brief historical overview, and then we look at putting principles into practice by reviewing the main FLR initiatives happening across each regional group.

There is no systematic compilation of FLR (or other forest restoration) projects globally, and even the Bonn Challenge has no mechanism for detailing what is proposed or instituted on the ground. Thus, our search went beyond 'FLR' to 'forest restoration' and 'forest rehabilitation'. This also recognizes that many projects labelled as FLR may not respond to the six globally agreed principles of FLR (or indeed, any of the definitions of FLR in use [21,41,45]). The Bonn Challenge Barometer is a limited attempt (12 countries to date, to be extended to 20 in 2020) to summarize efforts but provides only country-level results [46]. Some data from the 26 ROAM (Restoration Opportunities Assessment Method [47]) are presented in [48], but as they note, the assessments are owned by the governments and there is no requirement to make them publicly available. Thus, our results are limited rather than comprehensive, relying on compilations and case studies, mostly limited to sources in English. Data sources are summarized in table 1.

# 3. Background

## 3.1. Historical overview

Reversing degradation by restoring vegetation cover to degraded land has a long history [24], although the terminology of restoration is a relatively new development [71,72]. In 2000, a group of 30 social and natural scientists came together to define FLR as 'a planned process that aims to regain ecological integrity and enhance human wellbeing in deforested or degraded landscapes' [16,42,45]. This definition, together with associated research work and guidance [42,73–75], was to be the cornerstone of both WWF and IUCN's work on forest restoration in the next decades. Several years later, the Global Partnership on FLR (GPFLR) was established by WWF, IUCN and the UK Forestry Commission, which today regroups over 30 NGOs and private and public institutions. In 2011, IUCN joined forces with the German government to launch the Bonn Challenge on FLR, an attempt to achieve widespread political commitments towards the goal of restoring 150 million ha of forested landscapes by 2020; this was expanded by the New York Declaration on Forests to 350 million ha by 2030.

As of early 2020, nations, regions and companies have committed more than 173 million ha for restoration under the Bonn Challenge [11]. Responding to the Bonn Challenge also contributes to meeting national obligations under the several Rio Conventions [76]. For example, the CBD Aichi Target 15, UNFCCC REDD+ goals and the Rio+20 UNCCD land degradation neutrality targets are all intended to lead to carbon richer landscapes that are biodiverse, economically more productive, provide a sustained flow of a broad range of ecosystem services and are resilient to climatic variability [77,78]. FLR can contribute to several Sustainable Development Goals [7,8,26,79]. The current UN Decade on Ecosystem Restoration is the next chronological step in elevating the wider practice of ecosystem restoration to the forefront of international policy discourses [9].

In spite of the attention focused on FLR, consensus on what constitutes FLR and what differentiates it from functional or ecological restoration is elusive [17,80]. In line with better defining what counts as FLR, the GPFLR in 2018 proposed a series of six principles for FLR:

1. focus on landscapes;
2. engage stakeholders and support participatory governance;
3. restore multiple functions for multiple benefits;
4. maintain and enhance natural ecosystems within landscapes;
5. tailor to the local context using a variety of approaches; and
6. manage adaptively for long-term resilience [21].

Meeting all of these principles is ambitious, and in practice, many projects that have been carried out are being labelled as FLR while in practice, they do not qualify according to the definition of FLR (either

**Table 1.** Data sources used to compile information for analysis. Sources included online databases and websites, country reports and case studies.

| source | scope | comment | reference | link |
|---|---|---|---|---|
| Bonn Challenge | global | commitments by country and sub-country | [11] | https://www.bonnchallenge.org/commitments |
| Infoflr.org | global | summaries for 63 countries, including national restoration targets identified by an IUCN review | [49] | https://infoflr.org/countries |
| LAC 20×20 | Latin America and Caribbean | restoration projects | [14] | https://initiative20×20.org/restoration-projects |
| Restoration Database for Latin America and the Caribbean | Latin America and Caribbean | comparative research project on landscape restoration for emissions reductions, CIAT/WUR project for USAID | [50] | https://dataverse.harvard.edu/dataset.xhtml? persistentId=doi:10.7910/DVN/B90UOZ |
| AFRI 100 | Africa | data on the countries that have made commitments to the AFRI 100 initiative | [13] | https://afr100.org/content/countries |
| Regreening Africa | Africa | individual project reports | [51] | https://regreeningafrica.org/about/ |
| Forest Landscape Restoration Mechanism | global | FAO programme, country profiles | [52] | http://www.fao.org/in-action/forest-landscape-restoration-mechanism/background/en/ |
| FAO | Asia-Pacific | country overviews for China, Indonesia, Myanmar, Nepal, Philippines, Thailand and Vietnam | [53] | http://www.fao.org/3/a-i5412e.pdf |
| ROAM analysis | global | analysis of ROAM results; many individual country reports are not publicly available | [48] | |
| CFLRP projects | USA | collaborative Forest Landscape Restoration Projects, US Forest Service | [54] | https://www.fs.fed.us/restoration/CFLRP/results.shtml |
| CoRe | Parks Canada | Parks Canada restoration projects from across the country | [55–57] | https://www.pc.gc.ca/en/agence-agency/bib-lib/rapports-reports/core-2018 |
| ECCA30 | Caucasus and Central Asia | country summaries | [15] | https://www.unece.org/index.php?id=51698&L=0 |
| REACTION database | Northern Mediterranean | mostly early afforestation projects | [58] | http://185.23.121.66/wwwrestauracion/web/search.php |

(Continued.)

**Table 1.** (*Continued.*)

| source | scope | comment | reference | link |
|---|---|---|---|---|
| Restoration Opportunities Atlas, India | project descriptions | past and ongoing forest protection and landscape restoration initiatives in India | [59] | http://wri-sites.s3.amazonaws.com/ifmt/ROAManuals/Database%20on%20past%20and%20ongoing%20initiatives.pdf |
| SER | project descriptions | multiple countries | [60] | https://www.ser-rrc.org/project-database/ |
| SERA | Australasian case studies (Australia and New Zealand) | six case studies from Australia, two from New Zealand | [61] | https://www.seraustralasia.org/case-studies-1 |
| WWF | case studies | New Caledonia, Madagascar, Mexico, Lower Danube Landscape, Tanzania, WWF's worldwide field initiatives | [62–67] | https://wwf.panda.org/our_work/forests/forest_publications_news_and_reports/forest_reports/ |
| IUFRO | evaluation of case studies | lessons learned from case studies from Bangladesh, Brazil, Ethiopia, Ghana, Guatemala, India, Madagascar, Mongolia, Peru | [68] | https://www.iufro.org/uploads/media/op33.pdf |
| World Bank | land area | 2018 data on total land area, based on FAO statistics | [69] | https://data.worldbank.org/indicator/AG.LND.TOTL.K2 |
| World Bank | forest area | 2016 data on forest area as percentage of land area, based on FAO statistics | [70] | https://data.worldbank.org/indicator/AG.LND.FRST.ZS |

because of their scale or because they do not meet the dual ecological and social dimensions of FLR) or according to these principles.

## 3.2. Putting principles into practice

FLR is about returning some trees to a landscape so that they can enhance the overall benefits provided by forests in that landscape to both nature and people. This process takes place over a long period of time, large spatial scales and seeks to achieve multiple (social and ecological) objectives. Theoretically, a team of experts can design an ideal FLR scenario. This design may contain maps showing where different restoration activities might take place, it may contain an overall goal for the state of the future landscape and its management once restored, and it may contain a number of objectives for restoration at the level of individual sites [81]. Questions that arise from this first stage are: *Who develops this design? Who is involved in the design? Who leads and finances the design? Who is consulted in this design? How realistic is the design? Is there the knowledge and capacity in place to apply this design? What are competing designs?*

Implementation can take place once the FLR plan has been developed, in a stepwise approach (with some feedback loops) over a given period of time. Questions that arise from this stage are: *Is the timeframe for these actions realistic? Who coordinates them? With what authority? Who funds them? How long is the funding available? Is funding sufficient? How practical are these activities?* For instance, when projects are funded by external donors, the project timeframe may be unrealistically short (with donors typically funding projects for 3–5 years [82]) leading to small-scale interventions or often, to a lack of consultation simply to 'save time'.

Maps can identify where best to implement specific restoration activities. These may be derived from an 'ideal' optimal definition based on social and ecological conditions and indicators. Questions that arise are: *What was there in those sites before restoration? Why has land use changed? What are underlying drivers of the change? Who owns the land? Who owns the rights to access, use or manage the land? Are there conflicts over land use?* For example, while an optimal allocation of trees within the landscape may be defined to yield both social and ecological benefits, it is challenging to determine whose social (and economic) benefits (e.g. which community or local versus national stakeholders), and which ecological benefits should be prioritized (e.g. climate mitigation versus biodiversity conservation). Trade-offs are inevitable in practice and indeed, one positive aspect of FLR is that it is easier to make trade-offs involving contrasting activities/alternatives at a landscape scale than at an individual landholding [41].

## 3.3. Planning tools

In practice, there have been many challenges associated with implementing FLR programmes, not least reconciling the human and ecological dimensions and achieving the scale required. Several tools have been developed over the years to address some of these challenges (see notably [83] for an overview of decision-support tools for FLR). We review here three important planning tools that were designed to help implement FLR and to identify priority areas for implementation. Other tools for specific steps in the overall process can be found in [83] and [84].

### 3.3.1. A planning framework for FLR

The first attempt at outlining a process specifically for FLR was published by Vallauri *et al*. [85]. The intention through this framework was to provide indicative steps to planning a restoration initiative. Their process involved five steps:

— Step 1: Initiating an FLR programme—the purpose of this first step is to identify the problem(s) and agree on possible solutions and targets for restoration. In this step, stakeholders are engaged and consulted, and both social and ecological problems of deforestation and forest degradation are considered in order to identify ways of reversing them. The authors acknowledge that this first step could last several years.
— Step 2: Defining restoration needs and linking restoration to a large-scale conservation vision—in this step, an emphasis is on the biodiversity dimension of restoration with an explicit link being made to a wider 'conservation vision' for the area [86]. Potential benefits—social, ecological and economic—of restoration are assessed. This step leads to a definition of target sites for restoration within the landscape that are linked to the objectives identified.

— Step 3: Defining restoration strategies and tactics—this step looks at different trajectories or scenarios for achieving the objectives identified above. It acknowledges that trade-offs may be necessary and that reaching wider agreement among landscape stakeholders may take time and may require a phased approach. An important output anticipated at this stage is potential land use scenarios (including maps) and a fully costed restoration plan.

— Step 4: Implementing restoration—in this stage, the authors recommend starting small scale through pilot projects, emphasizing the need to 'learn by doing'.

— Step 5: Piloting systems towards fully restored ecosystems—this step advocates the need for long-term monitoring and adaptive management. It acknowledges that the plan identified earlier may need to be adapted based on feedback from the system, particularly given the complexity of working within a social–ecological landscape.

In practice, this guidance has been used for relatively small-scale projects. For example, in New Caledonia's dry forest, 10 partners have been collaborating to restore this remnant ecosystem since the early 2000s. Using guidance such as that of Vallauri *et al.* [85], the implementing partners mapped out priority restoration sites in a first phase of the project (2001–2006). Other planning frameworks have since been developed [87,88].

### 3.3.2. Guidelines for implementing FLR

In 2017, scientists from IUFRO collaborated on the development of guidelines for FLR implementation [17,81]. These followed a project management cycle and were split into four phases as follows:

— Phase 1: Visioning—this phase sets the goals, the purpose towards which an FLR project is directed. Visioning implies what a restored forest landscape might look like in a given context (country or landscape). In this phase, together with stakeholder consultation and a comprehensive situation analysis, describe expected long-term outcomes of FLR.

— Phase 2: Conceptualizing—the next phase provides concrete mid- to short-term targets, priorities, and social and ecological objectives.

— Phase 3: Acting—the acting or implementing phase transforms the overall objectives into concrete and measurable activities that will result in accomplishments or meet targets. It provides a sequenced list of what will be done, where, when, by whom and at what cost. This phase determines baselines and indicators of progress, enabling implementers to identify whether they are heading towards a successful outcome or not.

— Phase 4: Sustaining—this phase requires attention to the long-term, highlights planned interventions over time following a management plan, using adaptive management and monitoring that enables feedback loops so that changes to the plan may take place, as necessary based on subsequent developments.

The implementation guide has been translated into Spanish and French [89,90]. The guide has been used for training sessions with early career scientists from developing countries. More recently, the ITTO has published comprehensive guidelines that take the FLR principles and break them down into guiding elements [88] and NEPCon has released a draft field verification standard for ecosystem restoration [91].

### 3.3.3. An assessment of restoration opportunities

The first attempt at defining FLR opportunities was presented by IUCN and WRI in a 'World of Opportunity' map in 2011. In 2014, IUCN and WRI joined forces to develop the Restoration Opportunities Assessment Methodology [47]. Although not strictly about the entire process, this ROAM tool is highlighted here because of its widespread use and application that has an important influence on the way FLR ends up being implemented (or not) in different contexts. This methodology is aimed at defining and prioritizing opportunities and the course of action for FLR within a national or sub-national context (i.e. the visioning and conceptualizing phases of [17], based on an analysis of social, ecological and economic dimensions). This multi-factorial analysis is carried out in three broad phases:

— Phase 1: Preparation and planning—in this phase, a first analysis identifies the problem (of forest loss and/or degradation and underlying drivers) and proposes a broad overview of the target for restoration in the particular country context.

— Phase 2: Data collection and analysis—during this second phase, a series of analyses are carried out that form the backbone of the recommendations in the next phase. They are: a refined list of priority restoration interventions, based on a review of the initial interventions identified; a spatial analysis of restoration potential, including a series of national opportunity maps; an economic analysis of the costs and benefits associated with the identified restoration interventions; an analysis of the carbon sequestration potential and the associated co-benefits; a diagnosis of the presence of key success factors for restoration that examines the opportunities and challenges presented by the prevailing legal, institutional, policy, market, social and ecological conditions, as well as the implementation capacity and resources and the level of motivation among key actors; and an analysis of the financing and resourcing for the implementation of the identified FLR opportunities.

— Phase 3: Results to recommendations—based on the analyses and maps produced in the previous phases, this phase provides an opportunity to draft policy and institutional recommendations that lead up to the next implementation phase (beyond the ROAM).

Both Vallauri *et al*. [85] and Stanturf *et al*. [17,81] demonstrate similar basic steps in FLR, from a theoretical design, through to implementation (via pilot projects at times) and adaptive management based on feedback loops. The ROAM process [47] provides more detail on the first phase related to the design of an FLR programme, especially at the national level. Many other tools exist that are associated with ecological restoration [75,92], forest rehabilitation or specific elements of the overall restoration process [17].

# 4. Results: a global overview of the state of forest restoration

The need for ecosystem restoration is clear; an estimated 25% of the world's land area is degraded [93], threatening global sustainability. Deforestation, forest degradation, desertification, soil erosion, loss of productivity potential, biodiversity loss, water shortage and soil pollution are ongoing degradation processes. Responding to the adverse consequences of these processes requires a two-pronged approach: (i) avoiding or at least reducing degradation and (ii) restoring degraded ecosystems. The potential for restoring forest cover and functioning has been estimated variously as 1–2 billion ha globally [94–99], although the accuracy of these estimates has been challenged (e.g. [100–103]) and the strategy of tree planting for restoration or to mitigate climate change has drawn opposition, particularly in developing countries (e.g. [42,104,105]).

Although much of the Bonn Challenge discussion revolves around increasing forest cover by artificial or natural regeneration, many other restoration interventions are available (table 2). Multiple restoration objectives can be met by manipulating vegetation to increase structural complexity, change species composition and restore natural disturbance processes [107–110]. Livelihoods interventions are very context-specific but generally include alternative practices associated with the forest such as collecting or raising non-timber forest products (e.g. cane rats, medicinal plants, and honey), employment opportunities (e.g. collecting seeds of native trees, nursery work, planting and tending), and community forests and other interventions that address tenure and governance. Other interventions may indirectly relate to the restored forest landscape such as improved cook stoves that reduce the pressure on forests from fuelwood collection, climate-smart agriculture practices and improved seeds or livestock, agroforestry (including farm gardens with fruit trees, taungya and silvi-pasture) and developing value chains and access to markets for local products.

Tropical forests, which are the focus of many BC commitments, are extremely biodiverse, making them more difficult to restore when compared with temperate forests. Lamb [111] summarized the pros and cons of three restoration methods: natural regrowth, planted seedlings and direct seeding. Natural regrowth and direct seeding are low cost compared with planting seedlings; at the same time, planting is more reliable, especially for ensuring that preferred species, or those that do not disperse readily, are established. Several restoration methods have been developed specifically for tropical forest restoration to overcome some of these problems, including the framework species method [112,113] and maximum diversity planting [114] that plant mixtures of 20–50 species of different life forms and functional types [111].

## 4.1. Africa

The number of African countries making commitments to the Bonn Challenge far exceeds responses from countries on other continents (table 3). A regional initiative, AFRI 100, mostly overlaps with the Bonn

**Table 2.** Restoration methods to manipulate vegetation (based on [106]).

| methods | initial operations | comments |
|---|---|---|
| native recolonization | remove disturbance; fencing; re-establish hydrologic connectivity or physical processes for watershed, riparian, coastal restoration; stabilize site on mined land; leave alone if regeneration sources are available | variations include natural regeneration, assisted natural regeneration, farmer-assisted natural regeneration |
| afforestation, whole area | site preparation; plant or direct seed natives or non-natives as single rows or blocks | generally done in open land, often former agricultural land use; various objectives including watershed, riparian, or coastal restoration; species or landscape diversity. Reclamation may require physical alterations to stabilize spoils, chemical additions. For wood products, non-timber forest products, wildlife habitat, or carbon sequestration, planting or direct seeding may be natives, non-natives or naturalized non-natives |
| | site preparation; interplant; nurse crop; fast/slow-growing natives or non-natives; taungya | |
| | site preparation; plant complex mixtures of natives or non-natives; planting group method, framework species method; rainforestation | |
| afforestation, partial area | nucleation, cluster agroforestry | generally done in open land, often former agricultural land use; various objectives including watershed, riparian or coastal restoration; species or landscape diversity |
| afforestation, linear planting | site preparation; plant or direct seed natives or non-natives | generally done in open land, often former agricultural land use; riparian buffers, coastal barrier, dune stabilization; connecting forest fragments |
| conversion | clear fell and plant all desired species | used to restore degraded forests or as second intervention. Used in areas that were cleared or burned, lacking desired species; former swidden farming; blowdown, with or without salvage logging and planting desired species; agroforestry methods |
| | enrichment planting; framework species method | |
| | assisted natural regeneration; farmer-assisted natural regeneration | |
| | clear fell with residuals; variable density thinning | used to change stand structure or age structure at landscape level |
| | clear fell and plant | desired species and planting density will depend on objectives (carbon sequestration, species or landscape diversity; wood products, non-timber forest products or wildlife habitat) |
| transformation | partial overstorey removal | used to gradually change species composition, structure or both. Regeneration methods include natural regeneration, underplanting or enrichment planting, depending on timing of intervention. Could follow blowdown, with or without salvage logging |

(*Continued.*)

| methods | initial operations | comments |
|---|---|---|
| legacy retention or creation | deadwood, high stumps, artificial cavities, wounding | used to create species diversity in degraded forests lacking desired structure or as second intervention |
| reforestation for post-fire restoration | erosion control (reseed native understorey; mulching); with or without salvage logging; plant desired species | primarily for watershed restoration but could also be for wildlife habitat, wood products, carbon sequestration and other forest functions |
| reintroduce fire | fuel reduction by mechanical or chemical means; reintroduce prescribed fire; fire surrogates | restore fire regime and reduce risk large wildfires, increase species or landscape diversity |
| removal of invasive species | remove invasive species (hand clearing, mechanical, chemical); enhance natives (by controlling light, planting, etc.) | increase species or landscape diversity |
| replacement | stabilize site; plant seedlings of natives or non-natives; fertilize | used on highly disturbed sites (mined land, polluted land, avalanche track, landslide, lava flow). Various objectives including hydrologic functioning (watershed, riparian, coastal), geologic protection, species or landscape diversity |

Challenge, although there are differences, such as Mali and Sudan that have made large commitments to AFRI 100 but not to the BC. Only Burkina Faso and The Gambia have committed to the BC but not to AFRI 100 (table 3). In total, 31 African countries have committed to restoring between 94 and 126 million ha. Of these, nine countries have conducted ROAM assessments that identified three times more areas in need of restoration than were committed to the Bonn Challenge [48]. Agroforestry, reforestation and rehabilitation of degraded natural forests were the most identified interventions (table 4).

Two additional regional initiatives deserve mention, the African Great Green Wall of the Sahara and the Sahel and Regreening Africa. The Great Green Wall or Great Green Wall of the Sahara and the Sahel is led by the African Union; it attempts to reverse the effects of desertification by creating an 8000 km green barrier, a mosaic of green and productive landscapes across northern Africa. Rather than a massive tree planting programme, the Great Green Wall relies on farmer managed natural regeneration (FMNR), including the use of the traditional zai method of deep planting pits to enhance water infiltration and retention during dry periods. Stone barriers around fields contain runoff and increase infiltration from rain. By 2011, there were more than 4.8 million ha restored in Niger and more than 0.5 million ha in Mali [115–120]. Although not conceived as an FLR initiative, the Great Green Wall is landscape scale and increases trees in the landscape, thus Mali has made commitments to AFRI 100 but not the BC.

Regreening Africa is an initiative in sub-Saharan Africa to reverse land degradation on 1 million ha. It involves projects in eight countries (seven of which have made commitments under the Bonn Challenge; table 5) that seek to improve livelihoods and food security by integrating trees, crops and livestock using agroforestry techniques (table 5). Regreening Africa is funded by the European Union, managed by the World Agroforestry Centre (ICRAF), and implemented by partners including major international non-governmental organizations (INGOs), such as World Vision, Oxfam, Care International, Catholic Relief Services and Sahel Eco. Anticipated results are improved livelihoods for 500 000 households by increasing income, on average by 10% and environmental improvement by increasing tree cover by 10% and decreasing soil erosion by 5% [51].

Most FLR in Africa takes place in mosaic landscapes [95] where people and other land uses in addition to forests are significant. Thus, tenure security and governance issues are critical to successful restoration activities [39,121–125]. Case studies in Ghana and Madagascar [62,68], and ROAM

**Table 3.** Data on African countries that have made commitments to the Bonn Challenge, the AFRI 100 initiative, and restoration potentials or needs based on national assessments.

| country | land area (ha)[a] | forest cover (%)[b] | Bonn Challenge 2030 (ha)[c] | AFRI 100 (ha)[d] | national restoration needs/opportunities (ha)[e] |
|---|---|---|---|---|---|
| Benin | 11 276 000 | 38.2 | 500 000 | 500 000 | |
| Burkina Faso | 27 360 000 | 19.6 | 0 | 5 000 000 | 1 195 000 |
| Burundi | 2 568 000 | 10.7 | 2 000 000 | 2 000 000 | |
| Cameroon | 47 271 000 | 39.8 | 12 062 768 | 12 062 768 | |
| Central African Republic | 2 568 000 | 35.6 | 3 500 000 | 3 500 000 | |
| Chad | 128 400 000 | 3.9 | 5 000 000 | 1 400 000 | |
| Côte d'Ivoire | 31 800 000 | 32.7 | 5 000 000 | 5 000 000 | |
| Democratic Republic of Congo | 226 795 989 | 67.3 | 8 000 000 | 8 000 000 | 16 775 750 |
| Eswatini | 1 720 000 | 34.3 | 500 000 | 500 000 | |
| Ethiopia | 100 000 000 | 12.5 | 15 000 000 | 15 000 000 | 14 302 200 |
| The Gambia | 1 012 000 | 48.4 | 450 000 | 0 | |
| Ghana | 22 754 000 | 41.0 | 2 000 000 | 2 000 000 | 1 667 200 |
| Guinea | 24 571 000 | 25.9 | 2 000 000 | 2 000 000 | |
| Kenya | 56 914 000 | 7.8 | 5 100 000 | 5 100 000 | 4 210 000 |
| Liberia | 9 632 000 | 43.4 | 1 000 000 | 1 000 000 | |
| Madagascar | 58 180 000 | 21.4 | 4 000 000 | 4 000 000 | |
| Malawi | 9 428 000 | 33.4 | 4 500 000 | 4 500 000 | 7 700 000 |
| Mali | 122 019 000 | 3.8 | 0 | 10 000 000 | |
| Mozambique | 78 638 000 | 48.2 | 1 000 000 | 1 000 000 | 1 693 961 |
| Niger | 91 077 000 | 0.9 | 3 200 000 | 3 200 000 | |
| Nigeria | 91 077 000 | 7.2 | 4 000 000 | 4 000 000 | |
| Republic of Congo | 34 150 000 | 65.4 | 2 000 000 | 2 000 000 | |
| Rwanda | 2 467 000 | 19.5 | 2 000,000 | 2 000 000 | 1 585 030 |
| Senegal | 19 253 000 | 42.8 | 2 000 000 | 2 000 000 | |
| Sierra Leone | 72 180 000 | 43.1 | 0 | 700 000 | |
| South Africa | 121 309 000 | 7.6 | 0 | 3 600 000 | |
| Sudan | 0 | | 0 | 14 600 000 | |
| Tanzania | 88 580 000 | 51.6 | 5 200 000 | 5 200 000 | |
| Togo | 5 493 000 | 12.2 | 0 | 1 400 000 | |
| Uganda | 20 052 000 | 9.7 | 25 000 000 | 25 000 000 | 2 883 000/8 079 622 |
| Zambia | 24 193 000 | 65.2 | 0 | 0 | 1 596 700 |
| Zimbabwe | 38 685 000 | 35.5 | 2 000 000 | 2 000 000 | |
| total | 1 571 422 989 | | 117 012 768 | 148 262 768 | |

[a]2018 data, https://data.worldbank.org/indicator/AG.LND.TOTL.K2.

[b]2016 data, https://data.worldbank.org/indicator/AG.LND.FRST.ZS.

[c]Bonn Challenge website https://www.bonnchallenge.org/commitments.

[d]https://afr100.org/content/countries.

[e]https://infoflr.org/countries.

**Table 4.** Results from Restoration Opportunities Assessment Method (ROAM) reports from countries in Africa (source: [48]).

| national | scale | restoration interventions | BC (ha) | opportunity area (ha, identified in ROAM) |
|---|---|---|---|---|
| Burundi | sub-national (six provinces) | agroforestry, reforestation, rehabilitation of quarries, ecological agriculture, terracing, and plantations of bamboos and other appropriate indigenous plant species | 2 000 000 | 345 615 |
| Central African Republic | national | agroforestry, afforestation/reforestation, plantations (agricultural/forestry) and restocking of degraded areas | 3 500 000 | 7 650 000 |
| Côte d'Ivoire | national | agroforestry, reforestation and enrichment planting, and protection and natural regeneration of parks and reserves | 5 000 000 | 5 077 672 |
| Kenya | national | agroforestry reforestation and rehabilitation of degraded natural forests, woodlots, commercial tree and bamboo plantations; tree-based buffers along waterways, wetlands and roads; and silvopastoral and rangeland restoration | 5 100 000 | 38 800 000 |
| Madagascar | national | agroforestry, reforestation, restoration of degraded national forests, mangroves and pine forests | 4 000 000 | 11 122 540 |
| Malawi | national | agricultural technologies, forest management, soil and water conservation, community forests and woodlots, and river and stream bank restoration | 4 500 000 | 7 700 000 |
| Mozambique | sub-national (Zambézia and Nampula provinces) | agroforestry, forest restoration (natural regeneration, enhancement of existing forests and woodlands); water and watershed conservation, soil conservation and new forest plantations (planted forests and woodlots) | 1 000 000 | 1 639 961 |
| Rwanda | national | agroforestry, woodlot management, timber plantation management, forest restoration interventions and protective forests | 2 000 000 | 1 526 379 |
| Uganda | national | agroforestry, natural regeneration and woodlots | 2 500 000 | 8 079 622 |
| | | total | 27 600 002 | 81 941 789 |

**Table 5.** Data from countries with FLR projects funded under the Regreening Africa programme (source: [51]).

| country | region | area (ha) | households | restorative activities |
|---------|--------|-----------|------------|------------------------|
| Ethiopia | Northern, Central, Southern | 200 000 | 120 000 | FMNR, planting, agroforestry |
| Ghana | Upper East, Northern | 80 000 | 40 000 | FMNR |
| Kenya | Western | 150 000 | 50 000 | FMNR, tree-based value chains |
| Mali | Koutiala, Yorosso, Tominian, San | 160 000 | 80 000 | FMNR, tree-based value chains, grazing control |
| Niger | Simiri, Ouallam, Hamdallaye | 40 000 | 90 000 | agroforestry value chains |
| Rwanda | Eastern Savannah | 70 000 | 100 000 | FMNR, community nurseries |
| Senegal | Kaffrine, Kaolack, Fatick | 80 000 | 160 000 | FMNR, tree-based value chains |
| Somalia | Somaliland, Puntland | 20 000 | 40 000 | planting, tree-based value chains |

assessments in Uganda, Kenya and Tanzania [48], show how tenure security is crucial to FLR success. For example, the modified taungya method has been successfully applied in several projects in Ghana whereby landless farmers can grow crops in degraded forest reserves as native trees they planted mature, until trees shade out the food crops [126]. Additional livelihood benefits accrued included training in rearing the food delicacy cane rat (grass-cutter), the cultivation of different non-timber forest products (NTFPs), jobs to local communities through activities such as seedling production, tree planting and maintenance of plantations. Importantly, a benefit-sharing arrangement ensured improved tree tenure rights for farmers and local communities [127].

## 4.2. Latin America and the Caribbean

Deforestation, lowered productivity, aridity and water stress are common factors leading to land degradation in Latin America and the Caribbean [128]. Conversion to pasture and cropland are the major direct drivers of deforestation in South America [129] and logging is the most important driver of forest degradation [130]. Latin America and the Caribbean (LAC) 20×20 is a regional restoration initiative with many, but not all, of the national commitments also to the Bonn Challenge (table 6). Initiative 20×20 is a country-led effort to bring 20 million ha of land into restoration by 2020. So far, 17 Latin American and Caribbean countries and three regional programmes have committed to begin restoring 53 million ha of degraded land. The initiative is supported by more than 70 technical organizations and institutions and a coalition of impact investors and private funds deploying US$2.5 billion in private investment [14].

A recent assessment of 154 projects in Latin America and the Caribbean [131] included restoration projects funded by the Global Environmental Facility (GEF), Forest Investment Programme (FIP), the Clean Development Mechanism (CDM), as well as projects funded by governments and the private sector (LAC 20×20) and local efforts funded by NGOs, local governments and research organizations (figure 1). Their results highlight that most projects have similar visions and goals of increasing vegetation cover, recovering biodiversity and ecological processes; and many seek to improve livelihoods of local people, but funding source determined the types of activities pursued. For example, GEF and FIP projects favoured natural and assisted natural regeneration, while CDM projects favoured monoculture plantations, often of non-native species [131].

Coppus *et al.* [132] used a subset of 97 of these projects with complete information to categorize these projects according to size, amount of funding, funding sources and monitoring efforts. Three types emerged: (i) large-scale, well-funded by international donors, with established monitoring plans; (ii) local-level, privately funded projects mostly without monitoring plans; and (iii) local-level, low-cost, government-funded projects without monitoring plans. Funding source often determined the alignment of goals and activities carried out (table 7). Country and sub-country ROAM assessments provided more detail on planned interventions for some FLR projects (table 8).

### 4.2.1. Brazil

Two significant ecosystems in Brazil, the Atlantic Forest (Mata Atlântica) and the Amazon, have been heavily impacted by development, primarily for agriculture and pasture since European settlement.

**Table 6.** Data from countries in Latin America and the Caribbean that have made commitments to the Bonn Challenge and the LAC 20×20 Initiative, and restoration potentials or needs based on national assessments.

| country | land area (ha)[a] | forest cover (%)[b] | Bonn Challenge 2030[c] | LAC 20×20[d] | national restoration needs/opportunities[e] |
|---|---|---|---|---|---|
| Argentina | 273 669 000 | 9.8 | 1 000 000 | 1 000 000 | 1 000 000 |
| Brazil[c] | 835 814 000 | 58.9 | 12 000 000 | 22 000 000 | 3 200 000 |
| Chile | 74 353 200 | 24.3 | 500 000 | 500 000 | 600 000 |
| Colombia | 110 950 000 | 52.7 | 1 000 000 | 1 000 000 | 2 017 984 |
| Costa Rica | 34 150 000 | 54.6 | 1 000 000 | 1 000 000 | 234 347 |
| Ecuador | 24 836 000 | 50.2 | 500 000 | 500 000 | 0 |
| El Salvador | 2 072 000 | 12.6 | 1 000 000 | 1 000 000 | 1 000 000 |
| Guatemala | 10 716 000 | 32.7 | 1 200 000 | 1 200 000 | 825 026 |
| Honduras | 11 189 000 | 40.0 | 1 000 000 | 1 000 000 | 0 |
| Mexico | 194 395 000 | 33.9 | 6 500 000 | 8 468 284 | 10 475 077 |
| Nicaragua | 12 034 000 | 25.9 | 2 800 000 | 2 800 000 | 0 |
| Panama | 7 434 000 | 61.9 | 1 000 000 | 0 | 0 |
| Peru | 128 000 000 | 57.7 | 3 200 000 | 3 200 000 | 1 788 000 |
| | | | 32 700 000 | 43 668 284 | 21 140 434 |

| sub-country | Bonn Challenge 2030[ce] | LAC 20×20[d] |
|---|---|---|
| American Bird Conservation | | 100 000 |
| Bosques Modelo | | 1 600 000 |
| Brazil Espiritu Santu[f] | | 80 000 |
| Brazil Mata Atlântica[f] | 1 000 000 | |
| Brazil Mata Grosso[f] | | 2 900 000 |
| Brazil Sao Paulo[f] | | 300 000 |
| Conservacion Patagonica | | 1 000 000 |
| Guatemala Private | 40 000 | |
| Mexico Campeche | 400 000 | |
| Mexico Quintana Roo | 300 000 | |
| Mexico Yucatan | 250 000 | |

[a]2018 data, https://data.worldbank.org/indicator/AG.LND.TOTL.K2.

[b]2016 data, https://data.worldbank.org/indicator/AG.LND.FRST.ZS.

[c]Info FLR (IUCN) website https://infoflr.org/countries/.

[d]https://initiative20×20.org/regions-countries.

[e]https://infoflr.org/countries.

[f]Brazil is confusing. The country commitment under LAC 20×20 is 22 million ha, 12 million by the environment ministry and 10 million by the Ag ministry. It is unclear if the Mata Atlântica commitment of 1 million ha is included in the country commitment. Similarly, it is unclear if the state-level commitments are part of the national commitment under LAC 20×20.

The Atlantic Forest was settled first when colonizers reached Brazil and today is the focus of much of the restoration effort in Brazil. The Atlantic Forest originally extended for 130 million ha along the Atlantic coast of Brazil, Argentina and into Paraguay. Today only 7% of the forest in Brazil remains in good condition, distributed in isolated fragments. The Atlantic Forest Restoration Pact comprises over 270 signatory organizations working together to develop goals, priorities for research and monitoring plans; test and share results of innovative restoration methods [133]; and develop creative funding

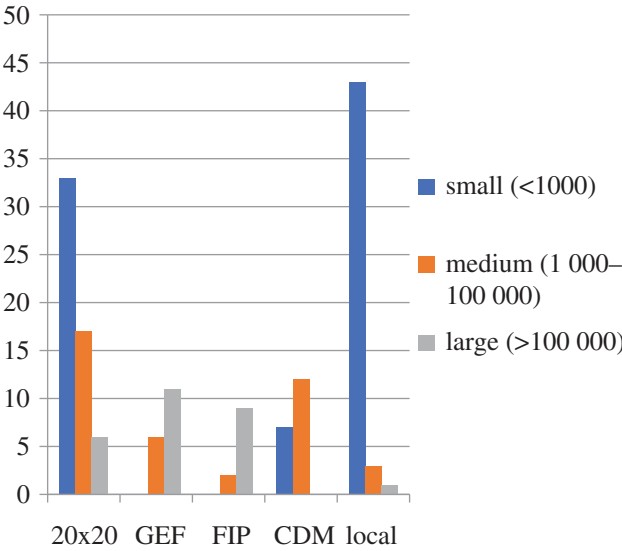

**Figure 1.** Number of FLR projects in Latin America and the Caribbean according to funding source and project size (source: [50]).

**Table 7.** Projects in Latin America and the Caribbean categorized according to size, funding level and source, and alignment with FLR goals and principles (source: [131,132]).

|  | Type 1 | Type 2 | Type 3 |
|---|---|---|---|
| size | >100 000 ha | 500–5000 ha | <100 ha |
| funding level | >US$10 million | variable | <US$500 000 |
| funding source | international donors | impact investors | national governments |
| monitoring | well-planned, with baseline | none | included for post-intervention assessments |
| goal alignment | national commitments to international agenda | environmental and socio-economic impact | environmental benefits |
| objectives | improving livelihoods, local capacity development, hydrological processes and water availability, climate change mitigation | economic revenue from timber and non-timber forest products | biodiversity improvement, regulation of hydrology |
| primary restorative activities | natural regeneration | mixed-tree plantations | excluding grazers |
| community involvement | low | low | moderate |

mechanisms to make restoration financially viable at large scales [134,135]. To date, 100 000 ha of restoration projects have been registered in the Pact and 1 million ha of secondary forests regenerated between 2008 and 2018. One of the main successes of the Pact is demonstrating that many individual restoration projects could be scaled up into a network of large-scale projects with a common objective [135]. Significantly, Brazilian law requires restoration, with specific criteria depending on the region and farm area, although enforcement has been spotty [136].

The Amazon continues to be subjected to deforestation and development, but some restoration has occurred; deforestation followed by conversion to agriculture and livestock have degraded large areas.

**Table 8.** Results from Restoration Opportunities Assessment Method (ROAM) reports from countries in Latin America (source: [48]).

| national | scale | restoration interventions | opportunity area (identified in ROAM) |
|---|---|---|---|
| Brazil | sub-national (two assessments: five states and Pernambuco separately) | agroforestry and restoration of native forest species | 3 937 722 ha (3 452 722 ha Pernambuco) |
| Colombia | national | biodiversity corridors for jaguars (*Panthera onca*) and puma (*Puma concolor*); a combination of good agricultural production practices, landscape restoration actions, and clear definitions of protected areas and buffer zones | 1 million ha Bonn Challenge |
| Costa Rica | national | silvopasture/improved pastures; passive regeneration; wood plantations; fertilizer management; crop residues management; agroforestry; of riparian forest | pastureland, coffee plantations and banana, palm oil and pineapple cropland (1 million ha Bonn Challenge) |
| El Salvador | national | landscape connectivity, carbon storage, fuelwood production, erosion improvement and nutrient delivery and mangrove restoration | 1 million Bonn Challenge |
| Mexico | sub-national (two assessments: Yucatán Peninsula states of Yucatán, Quintana Roo, and Campeche; Chiapas) | ecological restoration, rehabilitation of degraded forests, conservation agriculture, agroforestry systems, forest plantations and silvopastoral systems | over 2 million ha |
| Nicaragua | national | reforestation, natural and assisted regeneration, recovery of perennial crops, silvopasture systems, change in technologies and agroforestry | 1.2 million ha |
| Peru | sub-national (17 regions) | protected areas expansion, species migration corridors, reintroduction of habitats, reduce pressure on high-conservation value forests | 3.2 million ha Bonn Challenge |

Other factors of degradation, including fragmentation, fire, drought and invasive species, have exacerbated conditions. Mining companies in the 1980s began reclamation using native species [137]. This reclamation work has been costly, averaging US$2500 ha$^{-1}$ [138]. Another example of attempted restoration, the Xingu River basin in Mato Grosso state, is in the southeastern Amazon [60]. The aim is to restore and connect riparian forests in the Xingu watershed. After fencing to exclude cattle and cleaning to reduce fire risk, areas are planted using mechanized direct seeding of native species [139]. A selected mix of seeds of crops, fruits, green manure (annual and sub-perennial legumes) and native forest species is mixed with sandy soil. The mixture has at least three short-lived species (3 years); five species that live 30 years; 10 species that live 100 years and 15 long-lived species (greater than 100 years). The livelihoods component was the development of the Xingu Seed Network that

**Table 9.** Data on East and South Asian countries that have made commitments to the Bonn Challenge and restoration potentials or needs based on national assessments.

| country | land area (ha)[a] | forest cover (%)[b] | Bonn Challenge 2030 (ha)[c] | national restoration needs/potentials (ha)[d] |
|---|---|---|---|---|
| Bangladesh | 13 017 000 | 11.0 | 750 000 | 140 000 |
| Cambodia | 17 652 000 | 52.9 | 0 | 209 000 |
| China | 938 821 100 | 22.4 | 0 | 15 330 000 |
| India | 297 319 000 | 23.8 | 21 000 000 | 10 400 000 |
| Indonesia | 181 157 000 | 49.9 | 0 | 29 294 990 |
| Lao PDR (Sangthong district) | 23 080 000 | 82.1 | 0 | 52 985 |
| Myanmar | 65 308 000 | 43.6 | 0 | 1 200 000 |
| Mongolia | 156 412 000 | 08.0 | 600 000 | 0 |
| Pakistan | 77 088 000 | 01.9 | 100 000 | 1 755 982 |
| South Korea | 9 746 600 | 63.4 | 0 | 6 250 000 |
| Sri Lanka | 6 271 000 | 32.9 | 200 000 | 0 |
| Turkey | 76 963 000 | 15.4 | 0 | 30 000 |
| Vietnam | 31 007 000 | 48.1 | 0 | 17 235 554 |
| total | 1 902 108 700 | | 22 650 000 | 81 898 511 |
| *sub-national* | | | | |
| Asia Pulp and Paper | | | 1 000 000 | |
| Pakistan (Khyber Province) | | | 384 000 | |

[a]2018 data, https://data.worldbank.org/indicator/AG.LND.TOTL.K2.
[b]2016 data, https://data.worldbank.org/indicator/AG.LND.FRST.ZS.
[c]Bonn Challenge website https://www.bonnchallenge.org/commitments.
[d]https://infoflr.org/countries.

has produced and sold a large volume of seeds (over 175 tons) and generated about US$1 million for 450 households.

## 4.3. Asia

Five countries in East Asia (Bangladesh, India, Mongolia, Pakistan and Sri Lanka) have made formal commitments to the BC, totalling 22.65 million ha (table 9). Two sub-regional commitments by Asia Pulp and Paper in Indonesia and Kybher Pakhtunkhwa Province in Pakistan total 1.384 million ha. Four other East Asian countries have identified restoration potential/opportunities totalling 65 million ha (China, Indonesia, South Korea and Vietnam). China, the Philippines and South Korea have long histories of afforestation and reforestation [140–144] that pre-date the Bonn Challenge but have not made commitments to the BC.

### 4.3.1. Central Asia

No countries in Central Asia have made a formal BC commitment, although there are potentially almost 2.4 million ha suitable for restoration (table 10). Degradation drivers common to the countries in Central Asia are disturbances associated with steep terrain and seismicity [145]. Lowland areas have been cleared for agriculture and over-grazed; over-harvesting occurs in higher elevation areas [146]. In the boreal region, wildfires are an increasing problem [147,148]. In many countries in Central Asia, mineral extraction leaves mined areas in need of reclamation. Extraction of water for irrigation has reduced the areal extent of the Aral Sea, leaving a dry seabed of saline and sodic soils that are easily eroded [149]. The most pressing needs for restoration are near settlements, tugai or riparian forests, and mountain

**Table 10.** Data on countries in Central Asia, including potential Bonn Challenge commitments and restoration needs based on national assessments.

| country | land area (ha)[a] | forest cover (%)[b] | ECCA30 (ha)[c] | restoration needs[d] |
|---|---|---|---|---|
| Kazakhstan | 269 970 000 | 1.2 | 1 500 000 | agricultural conversion, dry bed Aral Sea |
| Kyrgyzstan | 191 800 000 | 3.3 | 323 000 | mining, over-harvest, grazing |
| Tajikistan | 13 879 000 | 3.0 | 70 000 | mining, over-harvest, grazing |
| Uzbekistan | 42 540 000 | 7.5 | 500 000 | agricultural conversion, dry bed Aral Sea, grazing |
| total | 518 189 000 | | 2 393 000 | |

[a]2018 data, https://data.worldbank.org/indicator/AG.LND.TOTL.K2.
[b]2016 data, https://data.worldbank.org/indicator/AG.LND.FRST.ZS.
[c]https://infoflr.org/bonn-challenge/regional-initiatives/ecca30.
[d][15].

forests that are degraded by fuelwood cutting, illegal logging and over-grazing [15]. Climate limits increasing forest area but agroforestry, in particular windbreaks near settlements and intensive agriculture could add trees to the landscape [15,150]. Nevertheless, FLR principles promoting stakeholder participation run counter to the remaining vestiges of Soviet-style central planning [15,151].

### 4.3.2. East and South Asia

Five countries in East and South Asia have made commitments to the BC, with the 21 million ha offered by India far over-shadowing the other four countries (table 9). The potential for restoration in other countries is great, particularly in China, Indonesia and Vietnam, although FLR is mostly a new concept. For example, forest rehabilitation and mining reclamation in Indonesia has been implemented at more than 400 locations since the 1960s, but few of these earlier projects produced positive results [152] and none can be considered FLR. A ROAM assessment identified restoration opportunities on 1.2 million ha in Myanmar (table 11). Vietnam's '5-Million Hectare Reforestation Programme' was launched in the late 1990s with the aim to establish 5 million ha of forest by the year 2010 (increasing forest cover from 28 to 43% by 2010), of which 1 million ha was to be through natural regeneration [153]. Many countries in Southeast Asia have attempted restoration of mangroves in coastal zones, especially in abandoned aquaculture ponds [154–156].

#### 4.3.2.1. India

Tree-based restoration activities in India have been implemented over the years by multiple actors in different regions of the country, from semi-arid to moist tropical biomes. Estimates of 'wasteland and degraded land' made by different agencies and criteria ranged from 46.7 to 187.7 million ha, although some of the areas could be desert or natural grassland [157]. As much as 39 million ha are thought to be suitable for bioenergy plantations [158], especially on saline [159] and sodic soils [160].

Recently, WRI India developed the Atlas of Forest Restoration Opportunities [161] to support the Bonn Challenge commitment of 21 million ha and India's nationally determined contribution (NDC) to the Paris Agreement. The Atlas identifies areas for protection and wide-scale and mosaic restoration. Wide-scale restoration potential was identified in areas where the dominant land use was forests, tree cover density was less than 40% (by definition open forest) and population density was less than 200 people per km$^2$. Mosaic restoration potential was identified on lands with less than 40% tree cover density and population density of less than 400 people per km$^2$; these included rainfed croplands. Wide-scale restoration was identified for 33.6 million ha, and 87.22 million ha as potentially suitable for mosaic restoration.

From 2011 until 2016–2017, a total of 9 810 944 ha were brought under restoration in India (figure 2). Most of the area was treated by government agencies (94%) under the National Afforestation Programme [162] which promotes participatory and sustainable management of degraded forests and adjoining areas. Assisted natural and artificial regeneration methods predominated, with lesser amounts of bamboo, mixed and silvi-pasture afforestation (figure 2). Lesser amounts of restoration were done by

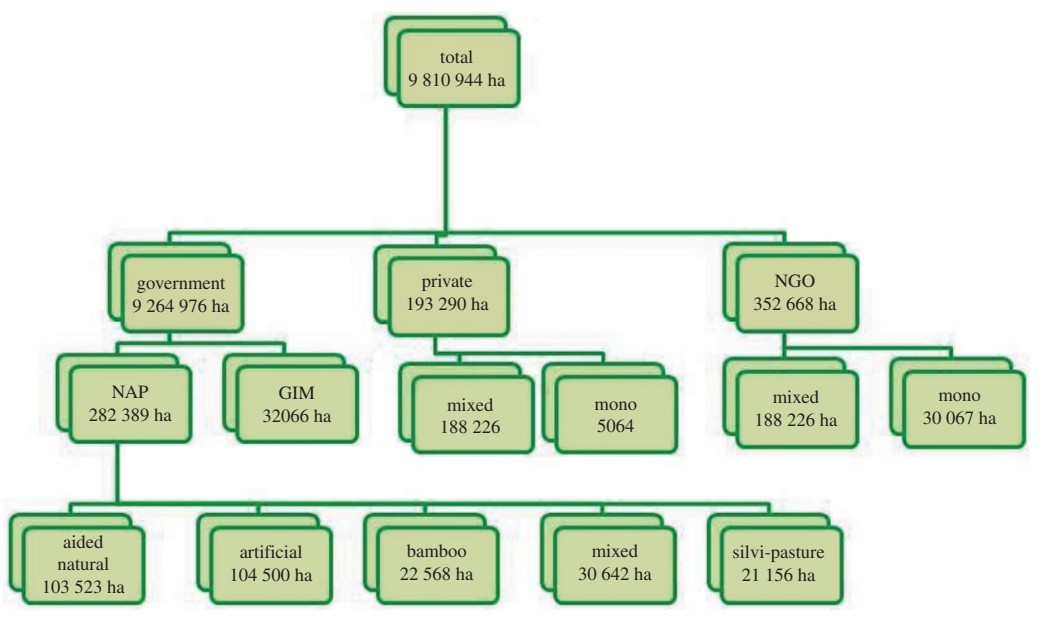

**Figure 2.** Plantation projects in India from 2011 to 2017 according to government, private industry or NGOs and by type of intervention (NAP, National Afforestation Program; GIM, Green India Mission) (source: [59]).

**Table 11.** Results from Restoration Opportunities Assessment Method (ROAM) reports from countries in Asia (source: [48]).

| national | scale | restoration interventions | opportunity area (identified in ROAM) |
|---|---|---|---|
| Cambodia | national | native forest restoration (timber, NTFP), assisted natural regeneration, tree planting, enrichment planting for locally extirpated species, flooded forest restoration with invasive species management and fire prevention | 209 000 ha |
| India | sub-national (Uttarakhand) | restoration strategies were designed for three distinct altitude zones within the state | 18.1% very high, 19.1% high restoration priority |
| Indonesia | sub-national (Sulawesi) | ecological mangrove restoration and hinterland agroforestry | 3018 ha |
| Lao PDR | sub-national (Sangthong district) | woodlots; natural regeneration; agroforestry and protection forest | 52 985 ha |
| Myanmar | national | natural regeneration; enrichment planting; improvement felling; climber cutting and thinning among natural regeneration | 1.2 million ha |
| Vietnam | sub-national (Quảng Trị Province) | enrichment planting/assisted natural regeneration, extended rotations, native species introduction, and soil and water conservation | 54 000 ha |

NGOs and private companies, mostly mixed plantations on smaller areas. The monocultures on private land were for commercial species such as rubber (*Hevea brasiliensis*); the mixed species plantations included non-native species such as *Leucaena* spp. and *Casurina* spp. The monocultures done by NGOs included mangrove plantings in coastal areas (*Avicennia* spp.) [162].

Many of the projects in India provided a variety of livelihoods benefits to local communities. In addition to organizing community forests and stakeholder dialogues, alternative livelihood activities included improved livestock rearing, kitchen gardens and fish for stocking local ponds. Employment opportunities were available for collecting seeds and wildings, planting, weeding and fire line cutting, and monitoring [59,162]. Enabling conditions were improved by changing legal status of forests that allowed communities to use and benefit from forests. Joint forest management schemes were initiated under various projects; most success was obtained when local communities formed effective committees such as Van Panchayat community forests in Uttarakhand [163].

## 4.4. Middle East and Mediterranean region

Restoration practice in the Mediterranean basin has evolved from revegetation to increasingly ecological approaches [164]. In the twentieth century, silvicultural approaches were used to combat erosion, protect watersheds and provide rural employment by relying on a few fast-growing tree species. Mostly, this led to single-species plantations, some of non-native species, with overall low diversity [165]. In the latter decades of the twentieth century, particularly in the European Union (northern Mediterranean countries), agricultural and biodiversity legislation has changed the focus to restoring native species or cultural ecosystems (e.g. oak woodland [166]).

A regional commitment to FLR in the Middle East and Mediterranean region was made at the 5th Mediterranean Forest Week, held in 2017, in Agadir, Morocco. This Agadir Commitment proposed restoring 8 million ha of degraded forest landscapes in the region by 2030. It was endorsed by 10 countries: Algeria, France, Iran, Israel, Lebanon, Morocco, Portugal, Spain, Tunisia and Turkey. This aspirational goal has to date not resulted in any formal commitments to the BC. Nevertheless, many of the signatory countries have a history of restoration, primarily afforestation of land degraded by grazing. Increasingly, wildfire is a concern as migrations from rural areas to urban centres in some countries have reduced active vegetation management exacerbated by drought [167,168] and likely to increase under future climate [169].

Lebanon and Morocco have not made formal country-level Bonn Challenge commitments as yet but are developing programmes with the assistance of the FAO programme, the Forest Landscape Restoration Mechanism (FLRM [52]). In 2014, the Ministry of Agriculture in Lebanon began the 40 Million Trees programme, a national afforestation/reforestation effort. The main aim is to increase forest cover from 13 to 20% by 2030; this will mean restoring forest cover to 70 000 ha. With assistance from the FLRM, a pilot project in the Shouf Biosphere Reserve (SBR) and in Kadisha Valley will restore abandoned agricultural stone terraces and plant them with various tree species. Similarly, in Morocco, a pilot programme in the Maâmora Forest in the Middle Atlas Mountains seeks to restore cedar ecosystems. The project will contract local users to maintain protected perimeters and develop alternative livelihood activities such as beekeeping or collection of medicinal plants in order to compensate for the temporary loss of grazing land. Nationally, a restoration target of 40 000 ha per year by 2020 has been included in the NDC of the Moroccan government. Ancient traditions in the region promote natural regeneration such as in Morocco, the Berber tradition of forest agdals or 'set asides' used in the High Atlas that allow the land to regenerate [170].

### 4.4.1. Turkey

Turkey is approximately 28.5% forested (2015 data), although not all forest lands meet the FAO definition. The forests are divided according to canopy closure: productive forests with 11–100% canopy closure constitute approximately 57% (127 000 km$^2$) of total forest area; degraded forests (porous), with 5–10% canopy closure account for 43% (97 000 km$^2$) of forest land. Total forest area increased slightly from 202 000 km$^2$ in 1972 to 223 000 km$^2$ as of 2015. Almost all forests are under government control; private forests account for less than 1%, approximately 18 000 ha [171]. Almost half of Turkey's forestlands are in need of restoration. Approximately 1090 ha of forests are illegally cleared annually for cropland [171]. Additionally, grazing impacts are still significant degradation threats in many regions. Fire-sensitive areas (125 000 km$^2$) are mostly in the Mediterranean Region; humans are responsible for 88% of fires [171].

Restoration in Turkey uses many of the techniques summarized in table 2, depending upon bioclimatic region (table 12). Afforestation efforts concentrated in arid and semi-arid regions can be perceived by local people in rural areas as a problem (e.g. elimination of their grazing

**Table 12.** National land degradation neutrality (LDN) goals for Turkey in forests (2015) (source: [171]).

| corrective measures | amount (ha) | cost (US$ million) |
| --- | --- | --- |
| afforestation | 600 000 | 900 |
| soil conservation afforestation | 900 000 | 630 |
| forestland rehabilitation | 1 500 000 | 450 |
| total | 3 000 000 | 1980 |

land) [172]. Afforestation in arid and semi-arid regions is primarily for erosion control. Some sites, however, are suitable for wood production [172]. Afforestation dates to Roman times when sand dunes were planted with stone pine (*Pinus pinea*) [172]. Modern afforestation progressed after World War II; by 1955, 4924 ha had been treated. By 2014, the Turkish Forest Service afforested 2.3 million ha for erosion control and 1.2 million ha for rehabilitation and restoration, mostly in semi-arid regions [172]. Future goals for Turkey are for restoration of 3 million ha by afforestation or rehabilitation [171].

Afforestation on sloping lands requires terracing and ripping and sometimes gully stabilization [172,173]. In the past, afforestation was mostly single-species plantations, established by planting bareroot seedlings or by direct seeding. Good results have been obtained by sowing *Quercus* spp. and *Cedrus libani*, broadcast by hand or from aeroplane. Increasingly multi-species plantations (polycultures) are preferred using native species. As a consequence, governmental nurseries have begun to produce broadleaved saplings instead of their past focus on coniferous species [173].

## 4.5. Europe and the Caucasus

Only two countries in Europe and the Caucasus have made Bonn Challenge commitments, Georgia and Scotland (table 13). According to its INDC submission to the UNFCCC in 2015 [174], Georgia will afforest/reforest 1500 ha of degraded lands and assist natural regeneration on 7500 ha to restore natural forest cover. Subject to available funding, Georgia will afforest/reforest up to a total of 35 000 ha and expand protected areas from 520 000 to 1 300 000 ha. Scotland committed to the BC early in 2019. It was a founding member of the GPFLR together with IUCN and WWF. Efforts to protect and reconnect the remaining fragments of the ancient Caledonian Forest dominated by Scots pine (*Pinus silvestris*) have been underway for decades. Recently, the Caledonian Forest rewilding effort has received an infusion of funding from the £23 million Endangered Landscapes Programme [175] that seeks to restore seven other major regeneration schemes in Europe.

There are numerous organizations in the UK advocating restoration of native woodlands and the focus has changed over time [176]. Once heavily forested (80% of the land area), by 1900, exploitation and agricultural conversion reduced forest cover to 5%. Beginning in 1920, government forestry programmes undertook large-scale afforestation, increasing forest cover to the current 12%. Because the afforestation was mostly on infertile sites, primarily non-native conifers were used with timber production objectives [163]. Current objectives have shifted towards broadleaves and multi-functional, diverse forests and these older plantations are being converted. Recently, the UK government announced the Woodland Carbon Guarantee (WCaG), a £50 million scheme to accelerate planting rates. The WCaG is a carbon market, rather than a grant or a fund mechanism that covers planting costs. Projects accepted into the WCaG have the option to sell carbon credits to the government every 5 or 10 years for a guaranteed price that is index-linked for the life of the contract, or the carbon credits can be sold on the open market at any time [177].

Three countries in Eastern Europe (Belarus, Moldova and Ukraine) have expressed interest in making commitments and assessed restoration potential and opportunity (table 13), although the distinction between restoration of degraded land and reforestation of harvested stands is blurred. Eight countries are EU members (Hungary, Romania, Poland, Czech Republic, Bulgaria, Slovakia, Croatia and Slovenia) and regard disturbances such as drought, pests and diseases as degradation drivers. Forest fires are a problem in Bulgaria, Croatia and Hungary; Croatia, Romania and Slovakia suffer wind damage [178]. Planting (afforestation and reforestation) are common, except in Bulgaria, which relies on natural regeneration, especially for converting conifer plantations to broadleaves. Hungary and

**Table 13.** Data on countries in Europe and the Caucasus, including potential Bonn Challenge commitments and restoration needs based on national assessments.

| country | land area (ha)[a] | forest cover (%)[b] | Bonn Challenge 2030 (ha)[c] | restoration needs/opportunities[d] |
|---|---|---|---|---|
| Armenia | 28 470 000 | 11.7 | | 50 000 |
| Azerbaijan | 82 670 000 | 14.1 | | 270 000 |
| Belarus | 20 291 000 | 42.6 | | 150 000 |
| France | 54 755 700 | 31.2 | | 10 371 240 |
| Georgia | 6 949 000 | 40.6 | 100 000 | 52 500 |
| Moldova | 3 287 000 | 12.6 | | 160 000 |
| The Netherlands | 3 369 000 | 11.2 | | 80 000 |
| Norway | 3 652 300 | 33.2 | | 1 000 000 |
| Spain | 49 956 400 | 36.9 | | 0 |
| Ukraine | 60 355 000 | 16.7 | | 8 526 600 |
| UK | 24 193 000 | 13.1 | | 1 596 700 |
| Scotland | 0 | | 170 000 | 0 |
| Russia | 1 637 687 000 | 49.8 | | 12 972 400 |
| total | 1 972 035 436 | | 270 000 | 35 229 440 |

[a]2018 data, https://data.worldbank.org/indicator/AG.LND.TOTL.K2.

[b]2016 data, https://data.worldbank.org/indicator/AG.LND.FRST.ZS.

[c]Bonn Challenge website https://www.bonnchallenge.org/commitments.

[d]https://infoflr.org/countries.

Poland have targeted increases in forest cover by planting; Bulgaria expects an increase of 40 000 ha by natural forest expansion and Romania plans 400 000 ha of afforestation by 2030 [178].

Four countries in the Balkans (Albania, Bosnia-Herzegovina and Montenegro, and the Republic of North Macedonia) face degradation from escaped agricultural fires. Windthrow is a problem in northern Serbia and landslides in the central region. Unpermitted logging and fuelwood harvesting cause over-exploitation in Serbia, also in Bosnia and Herzegovina and the Republic of North Macedonia. Bosnia-Herzegovina has the potential to make a major commitment to the Bonn Challenge; 1 million ha of coppice forests and 300 000 ha of brush and barren land have restoration potential. Albania, the Republic of North Macedonia and Serbia have potential to increase forest cover of, respectively, 10 000, 3000–5000 and 5000 ha yr$^{-1}$. Alternatively, Montenegro estimates 24 078 ha of improvement in stand quality rather than increase of forest area.

Similar to the Mediterranean Basin, forest restoration paradigms in northern Europe have shifted from the utilitarian to the idealistic. Even in the nineteenth century, afforestation of degraded heathlands aimed to revegetate to control erosion and provide rural communities with wood for fuel and construction as well as employment, sometimes overlain by an appeal to patriotism (e.g. [179]). Wood shortages after the two world wars drove afforestation with conifers and fast-growing *Populus*. In many countries, changing agricultural policies and EU incentives to afforest marginal farmland furthered the shift away from conifers to broadleaves, although targets for incentive programmes within the Common Agricultural Policy were lower than the initial expectations [180,181].

Afforestation programmes in Europe that developed after World War II were implemented at different rates; low productivity agricultural lands were abandoned, leading to afforestation or natural regeneration [182,183]. This was delayed in Eastern Europe until 1990 when the transformation from a socialist to a market economy created a similar condition [184,185]. In southern Europe, forest expansion was mostly due to natural colonization (spontaneous regeneration) (e.g. [186]).

By the late twentieth century, the shift to native species and more diverse landscapes was widespread. In northern Europe, a series of winter storms and widespread blowdowns illustrated the risk of off-site plantings of Norway spruce (*Picea abies*) [187–189] and furthered the rise of 'continuous cover' forestry that de-emphasized even-aged plantation management [190,191] and 'close to nature' silviculture with

emphases on natural regeneration, multi-species stands and increasing broadleaves over conifers [192]. The conversion of conifer stands to broadleaves has had to contend with the continuing positive economic returns of even-aged conifer plantations [193–196].

More recently, the concept of rewilding has taken hold in developed countries, whereby ecosystems are left to restore without human intervention, although in Europe, it often begins with a lot of intervention such as removal of agriculture and livestock, fencing exclosures, reversing drainage systems, etc. [197,198]. Originally rewilding focused on introducing keystone predators (e.g. wolves in North America [199]) and large vertebrates such as ungulates [200,201] and beaver in Europe [202,203].

## 4.6. North America

The two countries in North America, Canada and the USA (Mexico is included in Latin America and the Caribbean), rank third and fourth, respectively, for forest land area, accounting for 20% of the global total. Both are developed countries with significant industrial forest land, although they differ in land ownership; the USA has significantly more privately owned forest land (58%) than does Canada (6%). The USA has committed 15 million ha to the BC, but as yet, Canada has made no commitment [204]. Nevertheless, forest restoration is active in both countries.

### 4.6.1. USA

FLR in the USA occurs on both public and private land, with financing from public and private sources. The specific Bonn Challenge commitment of 15 million ha was made by the federal Forest Service, an agency within the Department of Agriculture, and takes place mostly on public land. Other federal restoration programmes, which are not part of the Bonn Challenge commitment, provide financial and technical assistance to private landowners in return for limited or perpetual conservation easements. Restoration programmes by private-sector actors with conservation and carbon management objectives also occur on privately owned land.

The Bonn Challenge commitment of the Forest Service is achieved though Collaborative Forest Landscape Restoration Program (CFLRP), primarily on public lands that are part of the National Forest System, involving state and local groups and tribal authorities (table 14). The CFLRP began in 2010 with 10 projects; another 13 started in 2013. Currently, most of the 23 landscape projects across the USA are in the western states because most land in the National Forest System is west of the central plains; only six of the projects are on forests in the eastern USA.

The landscapes and community groups involved in CFLRP are diverse, but the collaboratives faced some common challenges [205,206]. Not surprisingly, trust and the capacity to collaborate was one of the three broad categories of challenges, even among stakeholders without a contentious history. The other challenges were coming together to meet multiple objectives, and the ability to integrate ecological science and social values in decision-making. Previous studies have suggested that first addressing issues where there is consensus among stakeholders can build relationships and advance long-term goals [207]. In the collaboratives, focusing on improving ecosystem resilience has been effective in overcoming controversial topics such as thinning.

Funding for the CFLRP came from different sources: direct appropriation to the CFLRPs (27%), augmented by other Forest Service funds from the Washington Office and the national forest units involved (50%); and matched by funds from partners though agreements (7%), in-kind (11%) and goods and services provided (4%). Despite some data missing for some years, funding for this programme has been at least $915 million from the Forest Service and totalling over $1.2 billion over 10 years (table 14).

Two voluntary incentive programmes by federal agencies are aimed at private lands, the Conservation Reserve and Wetlands Reserve Programs (CRP and WRP). Both target fragile and marginal farmland with activities including tree planting. The CRP initially focused on highly erodible soils [208] and has evolved to include wildlife, water and air quality, and other conservation goals. For example, in addition to planting trees, other requirements have been added such as thinning to enhance wildlife habitat [209]. Farmers and ranchers enrol in CRP for 10- or 15-year contracts to maintain continuous cover in return for annual rental payments and cost-share and technical assistance. The CRP has enrolled 12.7 million ha; annual payments average US$21.45 ha$^{-1}$ at a total yearly cost of US$1.7 billion.

The WRP is aimed at another type of fragile lands, wetlands. Agricultural conversion accounted for 87% of wetland loss between the 1950s and 1970s, which the WRP was intended to reverse. The WRP is similar to CRP in structure, in that private landowners are offered financial incentives to take land out of active agriculture and restore to more natural conditions. Specifically, the WRP offers three types of

**Table 14.** Data from Collaborative Forest Landscape Restoration Program (CFLRP) of the United States Forest Service (source: [54]).

| project name | state(s) | size of the restoration landscape (ha) | cumulative footprint (ha) | land ownership patterns | goals | funding |
|---|---|---|---|---|---|---|
| Accelerating Longleaf Pine Restoration | Florida, Georgia | 229 786 | 191 685 | 41% Forest Service<br>24% Fish and Wildlife Service<br>15% private<br>13% industrial<br>7% state | accelerate ongoing longleaf pine (LLP) restoration by doubling the annual prescribed fire acreage to reduce fire return interval of 2–3 years associated with healthy LLP forests<br>reducing hazardous fuel loads using mulching/mastication to facilitate the reintroduction of prescribed fire into fire-suppressed areas<br>thinning small-diameter trees to reduce hazardous fuels, restore LLP dominance, and improve habitat for endangered and other wildlife species, improve conditions for diverse groundcover, and reduce the risk of insect and disease outbreaks<br>thin mature LLP stands, mixed longleaf/slash pine stands (selective slash removal) and convert slash pine plantations to LLP<br>restore groundcover in stands with high basal areas of small-diameter pines by thinning, chopping and burning on a 2–3 year rotation to stimulate grass and herbaceous ground cover to improve habitat for imperilled grassland birds such as Bachman's sparrow, Henslow's sparrow and bobwhite<br>decommission trails and roads for hydrological restoration on wet sites by blocking road access, planting container trees and shrubs, light discing to increase ground cover and/or recontour ditches and berms to restore normal hydrologic sheet flow | $30 710 813 |
| Amador Calaveras Cornerstone | California | 158 197 | 15 581 | 77% Forest Service<br>22% private<br><1% BLM and Bureau of Reclamation<br><1% state | restore and maintain functions of high-value watersheds<br>reduce wildfire in the wildland–urban intermix<br>promote aquatic and terrestrial health, biological diversity and habitat for native species, especially species at risk<br>create more resilient vegetation conditions<br>create markets for small-diameter trees<br>integrate ecological restoration with social goals, such as local employment and community social infrastructure development | $29 560 507 |

**Table 14.** (Continued.)

| project name | state(s) | size of the restoration landscape (ha) | cumulative footprint (ha) | land ownership patterns | goals | funding |
|---|---|---|---|---|---|---|
| Burney Hat Creek Basins | California | 149 347 | 15 188 | 54% Forest Service<br>7% national park<br>1% Bureau of Land Management<br><1% Bureau of Indian Affairs<br>37.5% private<br>7.4% industrial<br><1% state/local | re-establish a fire-adapted landscape<br>re-establish healthy forest conditions resistant to insects, disease and climate change<br>repair hydrologic function<br>restore and protect wildlife and fish habitat<br>sustain ecological refugia<br>provide sustainable use of forest products<br>enhance recreational opportunities and quality of experience<br>collaborate across land ownerships | $11 180 791 |
| Colorado Front Range | Colorado | 607 042 | 12 883 | 50% Forest Service<br>50% state and private | restore historic fire regimes (low-intensity fires) in lower montane ponderosa pine and mixed conifer forests<br>reduce the threat of high-severity wildfire and subsequent post-fire watershed damage<br>reduce suppression costs<br>decrease the density of ponderosa pine and Douglas fir, create a more diverse age structure<br>maximize ponderosa pine old growth<br>increase meadows, patchiness and herbaceous understorey<br>increase the resistance of the trees in the lower montane zone to mountain pine beetle | $72 347 093 |
| Deschutes Collaborative Forest Project | Oregon | 104 350 | 23 270 | 75% Forest Service<br>25% private lands | decrease the risk of high-intensity wildland fire behaviour by reducing and maintaining fuel loads in ponderosa pine and dry mixed conifer forests<br>reduce fire risk by thinning, mechanical fuels reduction and prescribed burning to reduce the risk of high-severity fire in WUI residential areas and drinking water source watersheds<br>accelerate development of late successional stand structure by thinning from below to create opening canopies and gaps<br>restore watershed functioning with stream channel restoration, wetland enhancement and establishment, riparian thinning, and road decommissioning and closure<br>improve aquatic habitat with fish passage enhancements<br>treat invasive plants | $30 246 820 |

**Table 14.** (*Continued.*)

| project name | state(s) | size of the restoration landscape (ha) | cumulative footprint (ha) | land ownership patterns | goals | funding |
|---|---|---|---|---|---|---|
| Dinkey Landscape | California | 62 323 | 14 535 | 84% Forest Service<br>16% private | restore frequent fire regimes characteristic of the Sierra Nevada<br>reduce stand densities and fuel loads by mechanical treatments and prescribed fire<br>improve growth of existing plantations to accelerate development of late seral conditions<br>restore watershed function and habitat for aquatic species<br>eradicate or control noxious and invasive plant species in advance of mechanical and fire treatments<br>develop stand structures resilient to changing regional climate conditions | $32 615 398 |
| Four Forest Restoration Initiative | Arizona | 971 267 | 324 816 | 94% Forest Service<br>6% other | re-establish a multi-scale mosaic of multi-aged stands dominated by old trees interspersed with regenerating trees and grassy openings<br>preserve large diameter trees; use a 16″ dbh cutting cap<br>reduce fire danger by thinning and removing hazardous fuels, particularly in the WUI<br>strategically place treatments to allow for increased use of prescribed fire and wildland fire<br>facilitate the re-creation of timber and biomass markets | $282 773 201 |
| Grandfather Restoration | North Carolina | 133 695 | 16 870 | 58% Forest Service<br>37% private<br>2% national park<br>2% state | rehabilitate pine and oak forests using prescribed fire, thinning and reintroduction of shortleaf pine<br>lower wildfire severity and fire suppression costs<br>improve species composition and structure removing white pine, red maple, yellow poplar and other mesophytic species through timber stand improvements, biomass thinning and timber sales<br>treat for non-native invasive plants<br>treat eastern and Carolina hemlock for hemlock woolly adelgid pest<br>restore hydrologic functions by stabilizing stream banks, reintroducing species, removing artificial fish barriers and enhancing streamside vegetation | $12 115 653 |

**Table 14.** (*Continued.*)

| project name | state(s) | size of the restoration landscape (ha) | cumulative footprint (ha) | land ownership patterns | goals | funding |
|---|---|---|---|---|---|---|
| Kootenai Valley Resource Initiative | Idaho | 323 756 | 22 184 | 61% federal<br>26% private<br>13% state<br><1% city and county | re-establish and maintain natural fire regimes to reduce the risk of unwanted wildland fire<br>increase prescribed fire to increase shrub diversity, forested vegetation types and openings which benefit grizzly bears and flammulated owls<br>improve forest structure, composition and habitat by hazardous fuels reduction, commercial thinning and harvesting; reforestation favouring western larch and white pine<br>increase hydrologic condition and improve stream channel connectivity of watersheds by culvert upgrades, fish passage replacement, and in-stream and habitat improvements<br>reduce wildlife disturbances by reducing motorized routes and decommissioning roads | $16 983 835 |
| Lakeview Stewardship | Oregon | 268 025 | 59 236 | 74% Forest Service<br>26% private | restore forest health and conditions that approximate historical species composition and stand ages by protecting large, fire-resistant, old-growth trees and thinning and removal of small-diameter trees<br>restore natural fire regimes by accelerated thinning and prescribed burning programme, focused on the dry, low-elevation ponderosa pine and mixed conifer forests<br>eliminate and control spread of noxious weeds, remove juniper<br>maintain and improve aquatic and riparian habitat by restoring native riparian vegetation (willows and aspen) to lower stream temperatures and sediment loads, remove barriers to fish passage, in-stream placement of large wood<br>minimize impacts on water quality and flow by decommissioning roads to reduce density, maintaining remaining roads by clearing brush and trees from the travel-way, ditch and culvert cleaning, slough slide removal, instal water bars, dips, and earthen berms and/or cross-ditches | $56 125 971 |

**Table 14.** (Continued.)

| project name | state(s) | size of the restoration landscape (ha) | cumulative footprint (ha) | land ownership patterns | goals | funding |
|---|---|---|---|---|---|---|
| Longleaf Pine Ecosystem Restoration and Hazardous Fuels Reduction | Mississippi | 154 593 | 151 356 | Forest Service | re-establish longleaf pine in stands that are currently growing loblolly or slash pine but have a soil type that is better suited for longleaf by harvesting off-site pine species, site preparation, planting longleaf seedlings, releasing seedlings from competing vegetation, and increasing native herbaceous seed capability<br>create more open stands and favourable conditions for grasses and forbs to grow using prescribed fire and thinning small-diameter midstorey trees to help prevent and suppress Southern pine beetle and other insect outbreaks<br>reduce hazardous fuels using herbicides and prescribed fire; common plant species (gallberry, yaupon holly, titi and wax myrtle) produce volatile oils that add to the extreme fire behaviour<br>control non-native invasive species using herbicides, focusing on eradicating cogongrass and kudzu<br>restore pitcher plant bogs by cutting, lopping, and scattering encroaching brush and undesirable woody species and remove mistakenly planted pine trees<br>improve watershed health and wildlife habitat by decommissioning and maintaining roads | $43 190 341 |
| Missouri Pine Oak Woodlands | Missouri | 139 907 | 46 888 | 36% Forest Service<br>36% state<br>20% private<br>6% national park<br>2% nature conservancy<br><1% NGO | restore shortleaf pine and oak bluestem woodland in a mosaic of age and structural classes through mechanical thinning, prescribed fire, and reintroduction of natural fire and sale of small-diameter biomass<br>restore the historic fire regime through accelerated thinning and prescribed burning to control undesirable woody understorey, increase the volume and diversity of ground flora and reduce accumulation of down, dead fuels<br>reintroduce wildlife and bird species (elk, Bachman's sparrow and brown-headed nuthatch) and endangered red cockaded woodpecker<br>improve forest health by converting and salvaging overabundant red oak to reduce the incidence of oak decline and reduce the spread of southern pine beetle by thinning and prescribed burning | $116 925 922 |

**Table 14.** (*Continued.*)

| project name | state(s) | size of the restoration landscape (ha) | cumulative footprint (ha) | land ownership patterns | goals | funding |
|---|---|---|---|---|---|---|
| Northeast Washington Forest Vision 20/20 | Washington | 370 815 | 50 342 | 54% Forest Service<br>16% tribal<br>5% state<br>23% private<br>1% BLM<br><1% Other Fed | restore fire-resistant and resilient late/old forest structure by thinning trees smaller than 21 inches dbh, plant created openings<br>reduce wildfire risk and fire management costs by thinning small trees, reducing fuel loads and ladder fuels and increasing fire breaks<br>restore watersheds by thinning and prescribed fire in riparian areas to recruit future large woody debris and stream banks to<br>restore aquatic habitat by relocating or maintaining recreational trails to reduce sediment to creeks and wetlands<br>restore upland wildlife habitats for ungulates and lynx<br>treat noxious and invasive weeds to slow their spread | $40 743 051 |
| Ozark Highlands | Arkansas | 139 374 | 88 180 | 76.5% Forest Service<br>5.8% national park service<br>6.3% state<br>11.5% private | restore fire regime by reducing fuel loads by thinning and understorey removal<br>restore oak and pine woodlands by thinning to reduce basal area and release oak regeneration and increase the vigour of mast-producing hardwoods<br>increase open forest and canebrakes<br>improve watershed conditions and reduce sedimentation by maintaining, closing and decommissioning roads<br>improve fish passage by replacing stream and river crossings<br>restore biodiversity in aquatic ecosystems by increasing large woody<br>enhance habitat and increase carrying capacity for elk | $34 901 637 |

(*Continued.*)

| project name | state(s) | size of the restoration landscape (ha) | cumulative footprint (ha) | land ownership patterns | goals | funding |
|---|---|---|---|---|---|---|
| Selway Middle Fork | Idaho | 566 572 | 102 328 | 94% federal<br>1% state<br>4% private<br><1% tribal | emulate natural disturbance using prescribed fire, mechanical treatments and natural fire<br>reduce fuels in order to minimize the risk of high-severity fire, especially in the wildland urban interface areas<br>reduce firefighting costs<br>improve forest resilience to insects, disease and wildfire by variable aged stands<br>maintain or promote forest structure of old growth conditions<br>improve wildlife habitat across summer, winter and transitional ranges; create openings to stimulate forage growth for ungulates; retain old forest structure to provide cover and security habitat<br>reduce sediment delivery to streams by improving road drainage decommissioning roads<br>replace culverts to restore aquatic habitat connectivity<br>treat noxious weeds, eliminate/contain new invasive species<br>create jobs and promote biomass facilities, low-impact harvest systems | $56 042 499 |
| Shortleaf Bluestem | Arkansas<br>Oklahoma | 141 029 | 129 502 | Forest Service<br>state<br>private | simulate natural disturbance patterns and return forests to more open woodland conditions by increasing prescribed fire and timber harvesting<br>develop and maintain forested linkages among mature forest habitats by treating fewer, larger areas rather than many smaller areas to reduce the total edge<br>reduce basal area by thinning, retaining old large trees and mast-producing hardwoods<br>regenerate stands by irregular seed tree and irregular shelterwood<br>remove off-site loblolly pine and retain mixtures of native pines and hardwoods | $32 700 445 |
| Southern Blues Restoration Coalition | Oregon | 279 532 | 110 069 | 79% Forest Service<br>3% Bureau of Land Management<br>18% private<br><1% state<br><1% tribal | restore natural fire regime and reduce size and severity of wildfires by thinning, slash treatments, biomass removal and underburning<br>create a mosaic of historic stand structures by removing small-diameter trees to maximize large tree retention<br>increase habitat for ungulates by reducing overstocking, increasing browse species, increase grasses and forbs by underburning, constructing fencing for resting rotation, remove encroaching conifers and juniper in meadows and shrub lands<br>protect riparian features by fencing, replanting native species, road removal and decommissioning, and juniper control<br>improve aquatic habitat by eliminating invasive common carp and controlling brook trout | $45 824 756 |

**Table 14.** (Continued.)

| project name | state(s) | size of the restoration landscape (ha) | cumulative footprint (ha) | land ownership patterns | goals | funding |
|---|---|---|---|---|---|---|
| Southwest Jemez Mountains | New Mexico | 84 986 | 22 206 | 93% Forest Service<br>4% private<br>3% tribal | reduce the risk of uncharacteristic wildfire by irregularly thinning stands, removing merchantable wood, preparing and burning slash<br>restore natural fire regimes of low-intensity surface burns in treated or open areas and mixed intensity burns in untreated stands<br>improve fish and wildlife habitat using in-stream structures, eliminate non-native fish and add native fish species access; reduce invasive plants using various methods; 90% in riparian areas<br>improve water quality and watershed functions with exclosure fences (or barriers) to limit cattle/elk access<br>rehabilitate bare soils, stabilize streambanks, reduce conifer encroachment, plant riparian vegetation, decommission roads and trails<br>use woody by-products | $61 880 966 |
| Southwestern Crown of the Continent | Montana | 586 807 | 62 265 | 59% Forest Service<br>11% other public<br>30% private | reduce risk of wildfire in the WUI by removing fuels<br>restore forest structure processes and resiliency, promote diversity, establish a mosaic pattern consistent with the historic mixed-severity fire regime<br>maximize retention of large trees, reintroduce low-severity and low-intensity fire to establish open stands<br>remove unnecessary roads<br>evaluate and adjust future desired conditions under predicted climate change<br>re-establish natural stream channels and riparian environments<br>remove barriers to fish migration<br>maximize the productive use of forest products | $94 554 810 |

(Continued.)

| project name | state(s) | size of the restoration landscape (ha) | cumulative footprint (ha) | land ownership patterns | goals | funding |
|---|---|---|---|---|---|---|
| Tapash Collaborative | Washington | 659 635 | 25 624 | 51% federal<br>15% state<br>10% private<br>24% tribal | increase proportion of dry and mesic forested landscape in a mosaic of variable size patches and gaps with large diameter and old trees dominant; retain large snags and fire-tolerant tree species<br>reduce the potential for uncharacteristic wildfire effects and fire suppression costs by mechanical thinning and burning to decrease surface fuel loading and disrupt large fire growth and reduce fire behaviour and severity<br>emphasize development of spotted owl habitat and other species including the northern goshawk<br>reduce adverse effects on stream flows, sediment regime and flood plain by decommissioning, stabilizing and resurfacing roads; relocate roads at risk from increased peak flows; convert system roads to trails; bridge motorized fords; remove and replace fish passage barriers<br>supply existing and attract new forest product processing infrastructure that facilitates ecologically based restoration and creates sustainable local employment | $21 795 963 |
| Uncompahgre Plateau | Colorado | 404 694 | 38 539 | 56% Forest Service<br>25% Bureau of Land Management<br>1% state<br>18% private | reintegrate and manage wildfire to reduce the risk of unnaturally severe or large crown fires<br>restore ecosystem structure, composition and function<br>preserve old or large trees while maintaining structural diversity and resilience<br>focus treatments on excess numbers of small young trees where this condition is inconsistent with historic range of variability<br>re-establish meadows and open parks and re-establish grasses, forbs and robust understorey communities<br>defer livestock grazing after treatment until the herbaceous layer has established<br>work with state wildlife agency to manage big game populations to manage herbivory effects on understoreys | $25 047 457 |
| Weiser Little Salmon Headwaters | Idaho | 323 310 | 76 892 | 64% Forest Service<br>7% Bureau of Land Management<br>4% state<br>25% private | restore ponderosa pine-dominated forests by thinning to reduce fuel loads and restore historic stand structure, composition and function; create clumpy distribution of large trees and openings and maintain large areas with dense canopies<br>rehabilitate plantations by fuel reduction and move them towards large tree habitat<br>reintroduce fire and reduce fuel loads in aspen stands by prescribed burning<br>control invasive weeds by spraying herbicides | $52 863 170 |

**Table 14.** (*Continued.*)

| project name | state(s) | size of the restoration landscape (ha) | cumulative footprint (ha) | land ownership patterns | goals | funding |
|---|---|---|---|---|---|---|
| Zuni Mountain | New Mexico | 84 986 | 22 663 | Forest Service | reduce risk of uncharacteristic wildfire, and re-establish natural fire regimes by mechanical treatments, brush | $33 896 689 |
| | | | | tribal | disposal and mastication, pile and broadcast burning | |
| | | | | Defence Dept. | manage natural fires in piñon-juniper, meadows, ponderosa pine, and small amounts of mixed conifers | |
| | | | | Bureau of Land | restore old-growth forest and other structural and compositional conditions representative of the historic variability | |
| | | | | Management | using a combination of pre-commercial thinning, uneven-aged stand restoration and meadow restoration | |
| | | | | private | improve fish and wildlife habitat, including endangered, threatened and sensitive species | |
| | | | | state | prevent, remediate, or control invasions of exotic species | |
| | | | | | contribute woody by-products for social and economic community benefits | |
| | | 6 944 028 | 1 623 102 | | | $1 235 027 788.00 |

contracts: (i) a permanent easement that pays 100% of the value of an easement and up to 100% of easement restoration costs; (ii) a 30-year easement that pays up to 75% of the value of an easement and up to 75% of easement restoration costs; and (iii) a cost-share agreement (up to 75% of restoration costs) to restore wetland functions and values without placing an easement on the enrolled hectares [210]. Since 1995, private landowners have voluntarily enrolled over 1 million ha into the WRP [210]. Easement payments are based on the income forgone by the landowner, thus varying by region, crop and productivity [211]. In one study in three states of the Lower Mississippi Alluvial Valley (LMAV) where the bulk of WRP easements occur, Jenkins *et al.* [211] estimated the average value for cropland of US$400 ha$^{-1}$ yr$^{-1}$. This included the value of crop production (US$309) and government subsidy payment (US$91). Afforestation costs in the LMAV average US$680–900 ha$^{-1}$. Because the easement payment is made as a lump sum in the first year of the WRP contract, a discounted present value (over a 30-year time horizon) of the combined income offset and the restoration costs for planting native tree species yields an annualized cost of US$455–468 ha$^{-1}$. Current potential returns from carbon markets and hunting leases provide only US$70 ha$^{-1}$ yr$^{-1}$. Nevertheless, adding to this the potential market value of US$1035 ha$^{-1}$ yr$^{-1}$ from emerging ecosystem markets for greenhouse gas (GHG) and nitrogen (N) mitigation as well as wildlife habitat provision, shows that benefits could be more than twice the restoration opportunity costs [211].

The number of private and public–private restoration efforts has increased, both for conservation and carbon management objectives. Some of the prominent programmes are through The Nature Conservancy, the National Wildlife Federation and the American Forest Foundation. Groups have organized around specific forest types to restore for example the Longleaf Pine Initiative [212] or regions such as the Lower Mississippi Alluvial Valley [213]. There are many other restoration projects for small areas; many of these can be called ecological restoration projects as the focus is on biodiversity and ecological integrity and would not be considered FLR as they are not at landscape scale and most do not have a livelihoods component (e.g. [214]).

### 4.6.2. Canada

Canada is 40% forested, containing 30% of the world's boreal forest and 9% of the global forest area [215]. Resource extraction is a significant portion of economic activity in Canada, including forest products, minerals, oil and natural gas [216]. Although Canada has not made a commitment to the Bonn Challenge, there are numerous restoration activities taking place in the country [204]. Recently, the federal government announced a natural climate solution, an initiative to plant 2 billion trees over the next 10 years [217]. This could result in 1–4 million ha of active restoration across the country, when compared with the 400 000 ha planted every year to regenerate timber harvesting in provincial forestlands.

The Parks Canada's Conservation and Restoration (CoRe) programme [55] is noteworthy for its scope and ambition. The CoRe programme attempts to restore healthy ecosystems, protect wildlife, tackle climate change and recover species at risk in national parks and national historic sites [55]. Half of all restoration projects conducted by this programme are in collaboration with Indigenous Communities or partners. The projects that address species-at-risk focus on protecting and recovering the species, including improving habitat. Restoring natural fire regimes is an objective in projects across the country, by reducing plant density and initiating prescribed burning.

Three national parks in eastern Canada, Terra Nova, Gros Morne and Cape Breton Highlands, face a common ecological challenge, namely, a decline in forest health caused by too many ungulates and too little fire, a consequence of past decisions. Many species have been adversely affected, for example, the Bicknell's thrush in Cape Breton Highlands National Park [56]. In some cases, European hunting and settlements extirpated moose and wolves; in other situations, moose were not present until introduced in the 1800s. Once (re)introduced, and lacking any natural predators, moose populations thrived to the point where today, moose browsing is so intense that trees do not regenerate, and some areas have turned into open fields. Another factor was aggressive wildfire suppression from the 1960s to the 1990s that further disrupted natural regeneration processes, leading to over-mature and weaker forests. This has resulted in spruce budworm outbreaks, a natural disturbance in boreal forests that kills mature trees, but high moose populations interfere with natural regeneration.

These parks are acting to remedy the situation by reducing moose populations through hunting to allow regeneration of hardwood species and balsam fir (*Abies balsamea*) seedlings. In Cape Breton Highlands National Park, this is being done in collaboration with the Indigenous Mi'kmaq [218]. In some areas where moose populations have been reduced, native tree seedlings are planted. In other

national parks, planting native species is done to convert monocultures. For example, in Prince Edward Island National Park, restoring Acadian forest species that includes a mixture of sugar maple (*Acer saccharum*), yellow birch (*Betula alleghaniensis*), red oak (*Quercus rubra*), eastern hemlock (*Tsuga canadensis*), red spruce (*Picea rubens*), eastern larch (*Larix laricina*) and white pine (*Pinus strobus*) is underway by cutting patches and thinning in white spruce (*Picea glauca*) monocultures to mimic natural disturbance such as wind storms and natural mortality. Where sources of appropriate species are lacking, a diversity of Acadian species sourced from local nurseries are planted [57].

# 5. Oceania

No nation in Oceania has made Bonn Challenge commitments; nevertheless, there have long been significant efforts in Australia and New Zealand to restore native forests and reverse land degradation. Large areas of Australia were converted by European settlers to pasture and cropland and native forests to non-native trees, resulting in loss of biodiversity and hydrological imbalances. Similarly, in New Zealand, native vegetation has been reduced and biodiversity lost by agricultural clearance, livestock grazing and introduction of other non-native mammals and plants.

## 5.1. Australia

Landscape-scale restoration has a long history in Australia, exemplified by community-based movements such as Landcare and local groups promoting conservation of specific birds or animals (e.g. Friends of Leadbeater's Possum [219]). The Landcare movement developed in response to land degradation and forest fragmentation focusing on community-based conservation [220,221]. Local groups began advocating for environmental issues in the 1950s and the national Landcare movement officially began in 1989. There are approximately 6000 Landcare, Coastcare, Bushcare and community environmental groups. The Landcare community is actively involved in 12 large (i.e. 700–3000 km) national connectivity initiatives and approximately 20 smaller scale (i.e. 50–200 km) initiatives [219]. These projects attempt to connect remaining habitat fragments of biologically defined regions or sub-regions comprised of core protected areas, buffer zones and compatible land use. In addition to protecting important ecological functions of these stepping stones [222], many of these groups recognize the important role of people in the landscape [219].

Ecological restoration gained prominence with initiation of the Society for Ecological Restoration Australasia (SERA) and the promulgation of standards of practice [223]. In addition to SERA, 12 partner organizations developed the standards over a lengthy consultation period: agencies, researchers and industry organizations and individual practitioners. Six case studies of ecological restoration in Australia and two in New Zealand are on the SERA website [61].

Rainforests in tropical and subtropical regions of Australia were converted to agriculture since European settlement. Between 1860 and the early twentieth century, extensive areas of lowland rainforests in subtropical Australia were cleared. Additionally, 40 000–50 000 ha were converted to plantations of native conifers between 1930 and 1990 [224]. Large areas of the remaining rainforests in mountain ranges are managed for conservation. Clearance in tropical regions is more recent, between 1900 and the 1950s. Most of the remaining rainforests in the region are now conserved in the Wet Tropics World Heritage Area [225]. Since the 1990s, many government-sponsored schemes have subsidized restoration of rainforests with a high level of community involvement. Many projects were small, less than 5 ha in area. Although this promoted community involvement and buy-in, they were expensive (US$14 000 ha$^{-1}$ or more) and insufficient to meet the need for large-scale restoration [226].

An evaluation of tropical and subtropical rainforest restoration from up to 2002 [226] showed that many individuals and organizations were involved; community-based efforts accounted for two-thirds of all projects. Mostly, this work has taken place since 1990 and ecological restoration of rainforest on cleared land is a more recent activity than either enhancement of existing remnants or planting rainforest trees for timber. Even though there were many individual projects, in aggregate the area of replanted rainforest was only about 1000 ha [226]. In the subtropics, replanting was about 1500 ha. Farm forestry or mixed purpose plantings in the tropics added 1500 ha and a few thousand hectares in the subtropics.

In the drier region of Western Australia, approximately 20 million ha of native *Eucalyptus* forests were cleared for cereal crops in the 1950s to the 1970s. Conversion to agriculture resulted in salinization of the landscape, wind erosion, biodiversity loss and hydrologic imbalance [227,228]. Replacing deep-rooted

forests with shallow-rooted annual crops and pasture plants increased recharge, resulting in groundwater rise accompanied by salt discharge [229]. Restoration has been proposed that would restore landscape water balances and stabilize areas already salinized, reduce wind erosion, and restore biodiversity [228,230]. Financing could come from carbon payments or payments for environmental services [228].

Reclamation of mined lands is another major restoration activity in Australia and the region [231,232]. The jarrah (*Eucalyptus marginata*) forests in Western Australia, for example, are being cleared, at the rate of 1000 ha yr$^{-1}$, for bauxite, gold and coal mining. Approximately 18 000 ha had been rehabilitated through 2007 primarily with native species [233–235]. Although research has greatly increased knowledge of how to effectively reclaim mined land, actual practice falls short of the potential [236]. Community groups are asking the mining sector and government for more accountability and to address the problems associated with the many abandoned mines and poorly reclaimed mined lands [236].

## 5.2. New Zealand

Forest restoration in New Zealand has many similarities to Australia, including restoration and protection of native forests [237], removal of invasive species [238] and development of connectivity corridors. For example, the Cape to City programme (http://capetocity.co.nz/) in Hawke's Bay is a wildlife restoration project supporting New Zealand's goal of eradicating invasive mammals (rats, possums and stoats) by 2050. This evidence-based approach to management is supported by extensive research [239]. Other large-scale efforts such as Reconnecting Northland [240] and Banks Peninsula Conservation Trust [241] also seek to eradicate pest mammals.

Two new private and public initiatives target restoring trees in pastoral farming areas. The Trees That Count initiative [242] is a non-government programme promoting planting of native trees, mostly on private land, for biodiversity conservation, carbon sequestration and soil erosion control. The initiative is a conservation charity that operates an online marketplace that matches funders with tree planters to help plant 200 million native trees. The New Zealand government in 2018 announced a plan to plant 1 billion trees over a decade [243]. Multiple government programmes contribute to the broader goal; a One Billion Trees Fund concentrates on a tree planting grants scheme that targets two-thirds of planting as indigenous species and a partnership fund that provides financial support for key activities that enable planting including research [244]. Several barriers were identified early in these programmes that are commonly encountered in large-scale planting. Landowners lack general information about tree growing, costs and benefits throughout the forestry life cycle, and the non-financial benefits of tree planting. In particular, information is often lacking about growing native species that are considered non-commercial. Land ownership and governance are complicated. In the New Zealand context, a particular Māori Whānau (a sort of extended family group) may hold rights too small for viable forestry, requiring negotiating agreements with neighbouring groups that are too complex to be worth the effort [244].

# 6. Discussion

The aspirational goals of the Bonn Challenge (BC) and related regional initiatives are laudable, as well as those of other mega-planting initiatives (e.g. https://www.trilliontreecampaign.org/). They have certainly captured the attention of policymakers, donor agencies and the general public. If realized, global land restoration and protection targets would have a significant impact; by one estimate, global tree cover would increase by 400 million ha and protect 28% of the terrestrial area with the highest values of both biodiversity and carbon storage [245]. Nevertheless, these gains would come at the expense of crop and pastureland at a time when there are increasing demands for agriculture and bioenergy [246–248]. In many instances, where tenure is unclear or contested, restoration may also be akin to land grabs [249,250]. Global commitments disconnected from local contexts are a recipe for disappointment [17,251].

## 6.1. Characterizing initiatives

Many countries have made FLR commitments without specifying what interventions will be used. Various assessments, however, give primacy to increasing forest cover using afforestation, natural regeneration or reforestation. In the drier regions of Africa, the Greening Africa projects prioritize farmer-managed natural regeneration. Rehabilitating existing stands by changing structure and/or species composition is more common in Europe and North America.

Of the 63 nations, regions and companies that have committed more than 173 million ha to the BC since early 2020 [11], most of the countries with commitments are in Africa [13], Latin America [13] and Asia [6]. Most countries made BC commitments larger than 2 million ha and many countries committed areas greater than their forest or agricultural areas [245]. Rwanda and Burundi, for example, pledged more than 75% of their land area (table 3). The countries with the largest commitments are India (21 million ha), Ethiopia, the USA (each 15 million ha) and Brazil (12 million ha). Each of these countries is large in area and has taken a different approach to the BC. India has reported 9.8 million ha under restoration, largely from plantations and agroforestry [162,252]. The BC commitment of the USA is primarily restoring natural fire regimes through thinning and prescribed burning, rather than expanding forest area; increased forest area is the target of new forests for only 4% of the nearly 15 million ha [245]. Brazil depends on natural regeneration in frontier regions, reporting 9.4 million ha under natural regeneration in the Amazon [253]. Ethiopia has not published accomplishments to date; however, many large-scale land restoration projects have been in the highlands of the Tigray and Amhara regions [254]. Assessments in the Tigray region suggest that excluding livestock would promote natural regeneration [255] and the most promising interventions in the Amhara region are medium to large-scale afforestation and reforestation, improved management of remnant high forests and sustainable woodland management [256].

The BC commitments do not fully correspond to those made to the regional initiatives or to other assessments of areas in need of or prioritized for restoration interventions. The AFRI 100 countries have made commitments of 148 million ha but only 117 million ha to the BC (table 3). The trend is the reverse in Latin America; the 13 BC countries have committed 33.7 million ha, but the 11 LAC 20×20 countries have committed only 21.7 million ha; Brazil accounts for most of the difference (table 6). In Asia, six countries have made BC commitments for 22.6 million ha, but other assessments of 12 countries have identified 65 million ha with restoration potential (table 9).

Of the many interventions contributing to FLR, afforestation has a long history in many countries that continues under the BC, for example, Turkey [171] and India [252]. Other well-known afforestation examples pre-date the current interest in FLR, for example, South Korea [142,143], Europe [179,257–259], China [141,260], Israel [261] and the USA [262,263]. The distinction between afforestation and reforestation is not always clear. Both terms refer to re-establishing forest cover where it has been lacking for some time. The distinction hinges on the time interval and whether another land use has intervened. Simply put, afforestation is planting trees to create a forest where one has not existed for some time before, and another land use such as row crops or pasture has intervened, generally for more than 20–50 years. Reforestation is planting trees to re-establish a forest after one has been removed by human or natural disturbance, without an intervening period in another land use, although a degraded, understocked condition may persist for some time. Simple examples are afforesting abandoned farmland versus reforesting after logging. The distinction becomes blurry where low-intensity subsistence farming (e.g. swidden) occurs or farmers encroach on degraded forest land as in Ghana [68]. Afforestation has become controversial because of misguided efforts to establish forests on sites that ecologically are better suited to grassland or other non-forested ecosystems (e.g. [20,100]). Although some have sought to avoid this negative association by using terms such as forestation or reafforestation, these only add more confusion. Forestation, for example, includes both afforestation and reforestation. We hold to the historical meaning with the understanding that afforestation should only be employed where ecologically appropriate.

Landscape approaches have been proposed as a means of tackling both social and ecological dimensions [264]. They provide a way of better integrating different stakeholders and different interests when it comes to land use planning and change. Landscapes also represent a spatial scale that is sub-national but beyond individual sites [265,266]. In countries with long-held traditions of forestry, however, the focus is on stand-level assessments, processes and monitoring with little awareness of landscape approaches. For example, in Eastern Europe and Central Asia, restoration and degradation are viewed in the light of damage from natural disturbances (including wind, fire, drought, insects and diseases) and potentially how these are intensified by climate change [178].

Several countries that are not part of the BC have taken a landscape approach. For example, connectivity projects in Australia and New Zealand attempt to connect forest fragments and rely on local support to implement the programme with public and private funding. Major afforestation projects in China may have increased forest cover (data in some cases are unreliable), but in the past have used non-native species and may have removed local farmers without providing them with alternatives [141,260,267,268], although there is increasing emphasis on using native species [269–271]. Other landscape approaches, such as the Great Green Wall in Africa, eventually may be part of BC

commitments of some of the countries, but so far these have not been solidified. Similarly, the Philippines has not made a BC commitment, but the country has a long history of establishing community forests, with mixed results [144].

Primacy to restoring ecological functioning or integrity is characteristic of BC and non-BC countries. The Parks Canada CoRe projects [55,204], for example, target conversion of monoculture plantations to multi-species stands of native trees as well as planting open areas and removing or reducing populations of herbivores that hinder natural regeneration. In many countries, interventions in protected areas take this ecological approach.

## 6.2. Outstanding issues

Measuring success in FLR is complex and, although some monitoring frameworks are being proposed [272–274], much remains to be done. Attempting to assess both social and ecological impacts requires indicators at different levels and across different spatial and temporal scales. In practice, the 'easiest' indicator which is predominantly used remains the numbers of trees or hectares planted. However, such an indicator does not demonstrate the persistence of the trees or any actual benefit (social or ecological) from those trees (e.g. [80]).

Implementing restoration is difficult due to technical complexities in many of the regions where BC commitments have been made [17]. The emphasis in many countries on using native species requires collecting seed from many species about which we do not know much (low densities, germination requirements especially for recalcitrant seeds) and nursery practices [231,275]. Multi-species stands will need multiple interventions over time because of interspecific competition, especially in the tropics where it occurs early in stand development [111]. Unless long-term management plans for ecologically sound and socially beneficial land use are developed and supported by monitoring, and resources are committed, tending needs probably will be overlooked.

Deforestation/degradation may be ongoing due to the needs of local populations to meet their food needs. Other barriers associated with tenure and governance are active areas of research [27,121,123,276] and often need broader attention and change in policy and law than can be addressed at the local or even landscape level. A lack of technical capacity [277–279] can be a significant barrier to FLR at all organizational levels (i.e. from policymakers and government agencies to local communities and households). Technical capacity is critical for up-scaling restoration interventions based on research or pilot projects to the landscape scale [280,281]. This has been apparent in the top-down BC commitment process where land area targets are made apparently overlooking local constraints [245]. Top-down assessments expect rapid change and projected outcomes are too often portrayed as immediately available, which is unrealistic [251].

The need to address drivers of forest loss and degradation remains a challenge in most cases, with the main direct drivers being similar across the planet: agricultural expansion, infrastructure development, mining and urbanization [36,130]. Top-down interventions often fail to address local challenges, such as those related to insecurity of tenure or marginalization of already vulnerable groups that ultimately may have significant impacts on the success of restoration [28,282,283]. Recognizing locally relevant techniques and adapting to local conditions may not be the most rapid way of achieving global targets, but it acknowledges local conditions and context and allows for flexible interpretation of FLR guidance.

Too narrow a focus on forests in landscapes has led some to interpret this to mean that FLR requires planting or regenerating new forest cover. Designing FLR as the right trees in the right place avoids some of the criticisms that have included mistaking lack of forest cover as degradation (e.g. where native grasslands occur [101,284]) or thinking that degraded areas are empty. An over-emphasis on planting to increase forest cover misses the contribution that trees outside of forests (windbreaks, agroforestry) can contribute to biodiversity and climate mitigation goals [285,286]. Emphasizing expanding forest cover through area goals [80] also omits the need to restore existing degraded forests, especially in countries with high forest cover [178,206].

## 6.3. How realistic are expectations that FLR will reach the Bonn Challenge commitments?

FLR underpins the BC and its regional offshoots (AFRI 100, LAC 20×20 and ECCA 30). The FLR approach pre-dates the BC and has four foundational aspects: (i) FLR is a planned process, (ii) FLR is integrative at the scale of landscapes, (iii) FLR focuses on landscapes where forests are a dominant feature, and (iv) FLR has dual (and presumably balanced) aims to regain ecological integrity and enhance human well-being [16,41]. The Bonn Challenge based on current commitments, however,

does not tell the whole FLR story at the global scale, as some countries with landscape-scale forest restoration efforts are not part of the BC or regional offshoots. Just as significantly, some BC commitments apparently include activities that, to some eyes, are not 'real' FLR as they do not incorporate the full foundational concepts [26,28].

'Real' FLR incorporates all the four foundational traits. A planned process implies a long-term vision and active intervention in well-defined, bounded areas. The natural human tendency to plan in stages and phases, however, is often shattered by the reality of having to 'muddle' through [287]. On the one hand, the scale implied in FLR and the complexity intrinsic in dealing with a social–ecological system requires planning [17], yet flexibility is necessary; regular reappraisals and modifications are more realistic than strict adherence to the original plan. On the other hand, 'laissez faire' approaches that then retrofit the label FLR may be preferred to 'interventionist' approaches that seek to 'direct' the effort without considering context where it is not always welcome and sometimes backfires. Laissez faire approaches, however, may lack accountability and risk long-term persistence. While a middle approach may be preferable, the multiple crises we are facing may not allow the time simply to muddle through [288–291].

Landscape approaches are advanced as superior to sectoral approaches that often result in conflicting and multiple demands for the same land resources [76,292]. Landscapes are large and complex, providing different habitats where diverse uses can be accommodated [41,266]. Despite calls for restoring ecological complexity [293], landscapes are socio-ecological systems that present particularly difficult conditions of dynamism and change. Experience from engineering suggests that all successful efforts at designing complex systems have started with small successful efforts, i.e. pilot or proof-of-concept projects. Experience with scaling up research to large-scale implementation seems to validate approaching FLR carefully and establishing an experience base [135,281].

Seeking to meet both social and ecological objectives is a strength of the FLR process but also a challenge. Different disciplines and expertise are required for each dimension and the complexity of operating within a social–ecological system signifies that often the focus tends to be on either the social (or economic) or the ecological system rather than balancing both [26,294]. Explicit as well as implicit biases towards the social or ecological dimensions of FLR potentially can be avoided by multi-disciplinary teams working together at all phases, from visioning to sustaining [81].

The barriers to realizing the potential of FLR are substantial; Fagan *et al.* [245] analysed commitments of Bonn Challenge countries on multiple indicators in three categories of feasibility of meeting the commitments, likelihood that restoration outcomes would persist and the effectiveness of governance. They concluded that if commitments were to be realized, significant land use changes would be required, substantially affecting the agricultural economy. Others have examined the likely persistence of restoration interventions based on attributes of stakeholders, environmental context and governance structure [295]. Not surprisingly, the attitudes of local stakeholders are important; without recognized, long-term benefits to local stakeholders, restoration is likely to be short-lived, especially if the main benefits are short term and disappear once donor support is removed. The ability of local stakeholders to control land use is another factor related to governance and tenure security; not to be overlooked is that restored land may become a new asset at risk of exploitation by elite capture [249]. Many areas targeted for restoration are available because they are degraded and probably pose challenges such as low fertility, draughtiness, etc. Therefore, the speed of recovery may be slow and areas under restoration may appear unused and at risk for encroachment [295,296].

# 7. Conclusion

Protecting and restoring forests is essential to meeting the Paris climate goals, conserving biodiversity and addressing food security and livelihood needs [297]. The FLR approach as it was designed initially, and as it was strengthened in 2018, through the agreement of six principles, provides an avenue to reach both ecological and social objectives. However, in practice, initiatives are still in their infancy when it comes to fully adhering to the objectives of this approach, and many initiatives that are labelled FLR would not qualify under its definition or principles. Forest landscapes have moved up in the political agenda; the Bonn Challenge and New York Declaration on Forests have set goals of bringing into restoration 350 million ha by 2030, fully supporting the UN Decade on Ecosystem Restoration (2021–2030). Nevertheless, setting ambitious area targets is insufficient for making real change towards more sustainable land use and functioning ecosystems [80]. The FLR process provides a long-term, multi-objective and large-scale means to implement international targets into on-the-ground interventions.

However, in practice, there are limited data on details of Bonn Challenge commitments or on progress towards accomplishments. Furthermore, significant large-scale restoration activity is undertaken outside of the BC, notably by the private sector. There are opportunities to learn from these large-scale initiatives but also to enhance alignment with the FLR approach. Nevertheless, implementing restoration is difficult due to ecological and socio-economic complexities in many of the regions where BC commitments have been made and the time that will be required for change to become evident. Without meeting the long-term needs of local stakeholders, restoration is likely to be short-lived, especially if the main benefits are short term and disappear once donor support is removed. Despite these challenges and the initial results thus far achieved, it is important to maintain the BC and NY Declaration on Forests as global restoration initiatives but increase their effectiveness by implementing enhanced pledging criteria and a more comprehensive and specific monitoring system.

Data accessibility. Data used are in the public domain; sources are listed in table 1.

Authors' contributions. Both authors contributed equally to this work, including conception and design, acquisition of data, analysis and interpretation of data, and drafting the article.

Competing interests. We have no competing interests.

Funding. The authors are supported by their respective organizations.

Acknowledgements. The article is an activity within the work of IUFRO Task Force Transforming Forest Landscapes for Future Climates and Human Well-being. We acknowledge the useful discussions and shared ideas of the Task Force members, and the support of IUFRO Deputy Director Michael Kleine. Reuben Coppus shared an early version of the Restoration Database for Latin America and the Caribbean: comparative research project on landscape restoration for emissions reductions [50] and Jürgen Blaser for sharing a draft of [178].

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
