## [Reviewer comments · Royal Society Open Science]

Review History

RSOS-201218.R0 (Original submission)

Review form: Reviewer 1

Is the manuscript scientifically sound in its present form?

No

Are the interpretations and conclusions justified by the results?

Yes

Is the language acceptable?

Yes

Do you have any ethical concerns with this paper?

No

Have you any concerns about statistical analyses in this paper?

No

Recommendation?

Major revision is needed (please make suggestions in comments)

Comments to the Author(s)

The authors compile information, largely based on Bonn Challenge initiatives and organizational information sources, on the state of the art of Forest Landscape Restoration around the world. This task is difficult as there is no clarity on how to define restoration activity as conforming to FLR beyond the labels used to describe it. Furthermore, it is possible for FLR to evolve from more focused projects that take on broader landscape-scale aspects over time; not all projects are planned as fully complete FLR implementation at once. In fact most do not take form that way. So the state of the art is not really possible to assess at a global scale, in terms of FLR"ness." And there are no clear criteria that are useful beyond knowing details of the specific interventions undertaken or planned and how stakeholders are engaged, as the number of trees planted or the number of hectares restored do not uniquely represent FLR.

These considerations are important to interpret the many tables included in the paper and compilations of data that are pulled together. What does all of this mean? I'm afraid that we still have a very poor understanding of where and how FLR is being implemented around the world, even with all of this information. So the title is a bit misleading. And the situation will not get any better without more clarity on how to define what is FLR on the ground. This means getting to specific criteria and indicators based on the principles of FLR.

Specific comments: (page #s refer to the Journal PDF page)

Page 3, line 21: Major restoration activity, is being conducted in the Philippines, which is not even mentioned here. This is not under Bonn Challenge, but their own initiatives, called the Philippines National Regreening Program. (more info provided in later comment).

Page 4, line 27: how does an intervention get the FLR label? What does this mean in practice?

Page 4, Line 50: and how to ensure that FLR covers at least three bases (and more) if there are no standards, criteria, or indicators? See Chazdon et al. (2020) for a discussion of principle, criteria, and indicator frameworks for FLR.

Page 5, Line 37: this history is also described here:

Laestadius, L., K. Buckingham, S. Maginnis, and C. Saint-Laurent. 2015. Back to Bonn and beyond: A history of forest landscape restoration and an outlook for the future. *Unasylva* 245 66:11-18.

also cite: Sabogal, C., C. Besacier, and D. McGuire. 2015. Forest and landscape restoration: concepts, approaches and challenges for implementation. *Unasylva* 245 66:3-10.

Page 6, line 10: The GPFLR also made a decision to use "Forest AND Landscape Restoration" as a deliberate step to include non-forest ecosystems in landscape restoration process. This is ignored in this review. It is not a trivial point. FLR is not just about restoring forests at the landscape scale and should not be conflated with landscape-scale reforestation or afforestation or ecological restoration of forests.

Page 6, line 10: but there is no system in place for identifying criteria and indicators of these principles to really assess whether FLR is being conducted. See Chazdon et al (2020).

Page 6, line 31-41: This is a very top-down, technocratic vision of FLR. What about the stakeholders who live in the landscape? Who are these experts? They may not really know the landscape or its history and human and natural resources available. The landscape context is critical for FLR planning and implementation. FLR can be based on an existing project or land uses rather than constructed de novo. It can evolve over time and doesn't have to be rolled out all at once. This is a very project and time-scale dependent approach that is unrealistic and does not address long-term goals or progressive changes over time.

Page 9, lines 33-39: The ROAM process requires extensive stakeholder engagement at national or subnational level. It is not a conceptual exercise. It is about forming coalitions and getting buy in from multiple sectors. But it is just the beginning and does not get to the implementation stage. It was designed for national level but is now being used to stimulate subnational assessments and implementation.

Page 9, heading for section 5. Forest restoration is not the same thing as Forest and Landscape Restoration.

Page 10, line 50: To be clear, these are not hectares of forest. They are ha of dryland regions where tree cover has increased including agroforestry parklands and enhancing crop production based on FMNR.

Page 11, lines 6-8: This is why it is forest and landscape restoration. Includes areas that are predominantly grassland ecosystems.

Page 12, line 42: Atlantic Forest. the latest assessment using high-resolution imagery shows 28% forest cover (Rezende et al. 2018). Much of the forest here is in small fragments.

Page 12, with regard to Brazil, should also mention that restoration is mandatory, with specific criteria depending on the region and farm area (Chaves et al. 2015, Brancalion et al. 2016)

Page 13, line 40: should mention here the Philippines National Regreening Program, not a part of the Bonn Challenge, but a significant commitment that has been renewed. (Gregorio and Herbohn 2018, Gregorio et al. 2020, von Kleist et al. 2020). Also FAO has extensive involvement in FLR in the Philippines: <http://www.fao.org/3/a-i5412e.pdf#page=135>

Page 22, line 52: please check this figure. I think two digits are missing here?

Page 25, line 18: It is unclear how these efforts are really based on FLR approach. This paper is conflating rather than clarifying these interventions.

Page 25, section on Successes and failures: relevant paper by the first author that is not cited and presents similar conclusions (Höhl et al. 2020).

Literature Cited:

Brancalion, P. H., L. C. Garcia, R. Loyola, R. R. Rodrigues, V. D. Pillar, and T. M. Lewinsohn. 2016. A critical analysis of the Native Vegetation Protection Law of Brazil (2012): updates and ongoing initiatives. *Natureza & Conservação* 14:1-15.

Chaves, R. B., G. Durigan, P. H. Brancalion, and J. Aronson. 2015. On the need of legal frameworks for assessing restoration projects success: new perspectives from São Paulo state (Brazil). *Restoration Ecology* 23:754-759.

Chazdon, R. L., V. Gutierrez, P. H. Brancalion, L. Laestadius, and M. R. Guariguata. 2020. Co-Creating Conceptual and Working Frameworks for Implementing Forest and Landscape Restoration Based on Core Principles. *Forests* 11:706.

Gregorio, N., J. Herbohn, R. Tripoli, and A. Pasa. 2020. A local initiative to achieve global forest and landscape restoration challenge – Lessons learned from a community-based forest restoration project in Biliran Province, Philippines. *Forests* 11:475 <https://doi.org/410.3390/f11040475>.

Gregorio, N., and J. Herbohn. 2018. Implementing the National Greening Program in the Philippines: lessons learned. *Current Conservation* 12:25-28.

Höhl, M., V. Ahimbisibwe, J. A. Stanturf, P. Elsasser, M. Kleine, and A. Bolte. 2020. Forest Landscape Restoration – What Generates Failure and Success? *Forests* 11:938.

Laestadius, L., K. Buckingham, S. Maginnis, and C. Saint-Laurent. 2015. Back to Bonn and beyond: A history of forest landscape restoration and an outlook for the future. *Unasylva* 245 66:11-18.

Rezende, C., F. Scarano, E. Assad, C. Joly, J. Metzger, B. Strassburg, M. Tabarelli, G. Fonseca, and R. Mittermeier. 2018. From hotspot to hopespot: An opportunity for the Brazilian Atlantic Forest. *Perspectives in ecology and conservation* 16:208-214.

Sabogal, C., C. Besacier, and D. McGuire. 2015. Forest and landscape restoration: concepts, approaches and challenges for implementation. *Unasylva* 245 66:3-10.

von Kleist, K., J. Herbohn, J. Baynes, and N. Gregorio. 2020. How improved governance can help achieve the biodiversity conservation goals of the Philippine National Greening Program. *Land Use Policy*:104312.

Review form: Reviewer 2

Is the manuscript scientifically sound in its present form?

No

Are the interpretations and conclusions justified by the results?

No

Is the language acceptable?

Yes

Do you have any ethical concerns with this paper?

No

Have you any concerns about statistical analyses in this paper?

No

Recommendation?

Reject

Comments to the Author(s)

The article provides an extensive assessment of “Forest Landscape Restoration” (“FLR”), but with a low level of novelty and bringing few, if any, critical science issues on the matter. I think that a formal literature review and/or metadata analysis goal should be clearly stated and an objective method of gathering information exposed. The “state of the art” in a science journal should be built on sound science.

In the summary, authors loosely state that tree planting is “inexpensive” for mitigating climate change, what is questionable, both because it is frequently expensive, and because it has limited contribution in mitigation of present World emissions.

Moreover, while the article deals with forests and “forest landscape restoration”, it should not ignore that planting forests will be welcome to reverse landscape degradation and biodiversity loss in former forested areas only, otherwise it will be a form of degradation. Maybe a bit more of context will help, including in this summary what is specifically called as “forest landscape restoration” and acknowledge its limitations in terms of both real contribution for changing climate reality and in which landscapes it is applicable.

Introduction apologetically praises planting trees and the FLR" without any critical considerations. The text is built to justify the need of the article being introduced relying in reputed misunderstandings and "lack of consensus", but fail to convince about which sound information and critical considerations about FLR is going to be presented in the following pages.

The rest of the article, not surprisingly, follow the agenda of "forest landscape restoration", minimizing criticism and presenting a wordy and biased description of how the "FLR" have been implemented around the World. There is no sound discussion based in science and few scientific publications being brought to truly question what "FLR" really is and what is its scientific basis. For instance, one of the main questions about FLR is the risk of threatening non-forest ecosystems with afforestation; in a 72-page document in only three instances grasslands are mentioned, and no discussion is done about it at all.

Given that, I think that authors could step back from the "FLR" agenda and provide a sound scientific review of what have been done under the umbrella of "FLR" and how science can inform the practice of FLR in order to advance with its objectives without harming other environmental and social goals.

Decision letter (RSOS-201218.R0)

Dear Dr Stanturf,

The Editors assigned to your paper RSOS-201218 "Forest Landscape Restoration: State of the Art" have now received comments from reviewers and would like you to revise the paper in accordance with the reviewer comments and any comments from the Editors. Please note this decision does not guarantee eventual acceptance.

Please submit your revised manuscript and required files (see below) no later than 21 days from today's (ie 29-Oct-2020) date. Note: the ScholarOne system will 'lock' if submission of the revision is attempted 21 or more days after the deadline. If you do not think you will be able to meet this deadline please contact the editorial office immediately.

on behalf of Dr Agnieszka Latawiec (Associate and Subject Editor)
openscience@royalsociety.org

Subject Editor Comments to Author (Dr Agnieszka Latawiec):
Comments to the Author:

Dear Authors,

Please carefully revise your manuscript with respect to both reviewers that made a list of important considerations to be included.

Kind Regards,
Agnieszka Latawiec

Reviewer comments to Author:
Reviewer: 1

Comments to the Author(s)

The authors compile information, largely based on Bonn Challenge initiatives and organizational information sources, on the state of the art of Forest Landscape Restoration around the world. This task is difficult as there is no clarity on how to define restoration activity as conforming to FLR beyond the labels used to describe it. Furthermore, it is possible for FLR to evolve from more focused projects that take on broader landscape-scale aspects over time; not all projects are planned as fully complete FLR implementation at once. In fact most do not take form that way. So the state of the art is not really possible to assess at a global scale, in terms of FLR"ness." And there are no clear criteria that are useful beyond knowing details of the specific interventions undertaken or planned and how stakeholders are engaged, as the number of trees planted or the number of hectares restored do not uniquely represent FLR.

These considerations are important to interpret the many tables included in the paper and compilations of data that are pulled together. What does all of this mean? I'm afraid that we still have a very poor understanding of where and how FLR is being implemented around the world, even with all of this information. So the title is a bit misleading. And the situation will not get any better without more clarity on how to define what is FLR on the ground. This means getting to specific criteria and indicators based on the principles of FLR.

Specific comments: (page #s refer to the Journal PDF page)

Page 3, line 21: Major restoration activity, is being conducted in the Philippines, which is not even mentioned here. This is not under Bonn Challenge, but their own initiatives, called the Philippines National Regreening Program. (more info provided in later comment).

Page 4, line 27: how does an intervention get the FLR label? What does this mean in practice?

Page 4, Line 50: and how to ensure that FLR covers at least three bases (and more) if there are no standards, criteria, or indicators? See Chazdon et al. (2020) for a discussion of principle, criteria, and indicator frameworks for FLR.

Page 5, Line 37: this history is also described here:

Laestadius, L., K. Buckingham, S. Maginnis, and C. Saint-Laurent. 2015. Back to Bonn and beyond: A history of forest landscape restoration and an outlook for the future. *Unasylva* 245 66:11-18.

also cite: Sabogal, C., C. Besacier, and D. McGuire. 2015. Forest and landscape restoration: concepts, approaches and challenges for implementation. *Unasylva* 245 66:3-10.

Page 6, line 10: The GPFLR also made a decision to use "Forest AND Landscape Restoration" as a deliberate step to include non-forest ecosystems in landscape restoration process. This is ignored in this review. It is not a trivial point. FLR is not just about restoring forests at the landscape scale and should not be conflated with landscape-scale reforestation or afforestation or ecological restoration of forests.

Page 6, line 10: but there is no system in place for identifying criteria and indicators of these principles to really assess whether FLR is being conducted. See Chazdon et al (2020).

Page 6, line 31-41: This is a very top-down, technocratic vision of FLR. What about the stakeholders who live in the landscape? Who are these experts? They may not really know the landscape or its history and human and natural resources available. The landscape context is critical for FLR planning and implementation. FLR can be based on an existing project or land uses rather than constructed de novo. It can evolve over time and doesn't have to be rolled out all at once. This is a very project and time-scale dependent approach that is unrealistic and does not address long-term goals or progressive changes over time.

Page 9, lines 33-39: The ROAM process requires extensive stakeholder engagement at national or subnational level. It is not a conceptual exercise. It is about forming coalitions and getting buy in from multiple sectors. But it is just the beginning and does not get to the implementation stage. It was designed for national level but is now being used to stimulate subnational assessments and implementation.

Page 9, heading for section 5. Forest restoration is not the same thing as Forest and Landscape Restoration.

Page 10, line 50: To be clear, these are not hectares of forest. They are ha of dryland regions where tree cover has increased including agroforestry parklands and enhancing crop production based on FMNR.

Page 11, lines 6-8: This is why it is forest and landscape restoration. Includes areas that are predominantly grassland ecosystems.

Page 12, line 42: Atlantic Forest. the latest assessment using high-resolution imagery shows 28% forest cover (Rezende et al. 2018). Much of the forest here is in small fragments.

Page 12, with regard to Brazil, should also mention that restoration is mandatory, with specific criteria depending on the region and farm area (Chaves et al. 2015, Brancalion et al. 2016)

Page 13, line 40: should mention here the Philippines National Regreening Program, not a part of the Bonn Challenge, but a significant commitment that has been renewed. (Gregorio and Herbohn 2018, Gregorio et al. 2020, von Kleist et al. 2020). Also FAO has extensive involvement in FLR in the Philippines: <http://www.fao.org/3/a-i5412e.pdf#page=135>

Page 22, line 52: please check this figure. I think two digits are missing here?

Page 25, line 18: It is unclear how these efforts are really based on FLR approach. This paper is conflating rather than clarifying these interventions.

Page 25, section on Successes and failures: relevant paper by the first author that is not cited and presents similar conclusions (Höhl et al. 2020).

Literature Cited:

Brancalion, P. H., L. C. Garcia, R. Loyola, R. R. Rodrigues, V. D. Pillar, and T. M. Lewinsohn. 2016. A critical analysis of the Native Vegetation Protection Law of Brazil (2012): updates and ongoing initiatives. *Natureza & Conservação* 14:1-15.

Chaves, R. B., G. Durigan, P. H. Brancalion, and J. Aronson. 2015. On the need of legal frameworks for assessing restoration projects success: new perspectives from São Paulo state (Brazil). *Restoration Ecology* 23:754-759.

Chazdon, R. L., V. Gutierrez, P. H. Brancalion, L. Laestadius, and M. R. Guariguata. 2020. Co-Creating Conceptual and Working Frameworks for Implementing Forest and Landscape Restoration Based on Core Principles. *Forests* 11:706.

Gregorio, N., J. Herbohn, R. Tripoli, and A. Pasa. 2020. A local initiative to achieve global forest and landscape restoration challenge – Lessons learned from a community-based forest restoration project in Biliran Province, Philippines. *Forests* 11:475 <https://doi.org/410.3390/f11040475>.

Gregorio, N., and J. Herbohn. 2018. Implementing the National Greening Program in the Philippines: lessons learned. *Current Conservation* 12:25-28.

Höhl, M., V. Ahimbisibwe, J. A. Stanturf, P. Elsasser, M. Kleine, and A. Bolte. 2020. Forest Landscape Restoration – What Generates Failure and Success? *Forests* 11:938.

Laestadius, L., K. Buckingham, S. Maginnis, and C. Saint-Laurent. 2015. Back to Bonn and beyond: A history of forest landscape restoration and an outlook for the future. *Unasylva* 245 66:11-18.

Rezende, C., F. Scarano, E. Assad, C. Joly, J. Metzger, B. Strassburg, M. Tabarelli, G. Fonseca, and R. Mittermeier. 2018. From hotspot to hopespot: An opportunity for the Brazilian Atlantic Forest. *Perspectives in ecology and conservation* 16:208-214.

Sabogal, C., C. Besacier, and D. McGuire. 2015. Forest and landscape restoration: concepts, approaches and challenges for implementation. *Unasylva* 245 66:3-10.

von Kleist, K., J. Herbohn, J. Baynes, and N. Gregorio. 2020. How improved governance can help achieve the biodiversity conservation goals of the Philippine National Greening Program. *Land Use Policy*:104312.

Reviewer: 2

Comments to the Author(s)

The article provides an extensive assessment of “Forest Landscape Restoration” (“FLR”), but with a low level of novelty and bringing few, if any, critical science issues on the matter. I think that a formal literature review and/or metadata analysis goal should be clearly stated and an objective method of gathering information exposed. The “state of the art” in a science journal should be built on sound science.

In the summary, authors loosely state that tree planting is “inexpensive” for mitigating climate change, what is questionable, both because it is frequently expensive, and because it has limited contribution in mitigation of present World emissions.

Moreover, while the article deals with forests and “forest landscape restoration”, it should not ignore that planting forests will be welcome to reverse landscape degradation and biodiversity loss in former forested areas only, otherwise it will be a form of degradation. Maybe a bit more of

context will help, including in this summary what is specifically called as “forest landscape restoration” and acknowledge its limitations in terms of both real contribution for changing climate reality and in which landscapes it is applicable.

Introduction apologetically praises planting trees and the FLR” without any critical considerations. The text is built to justify the need of the article being introduced relying in reputed misunderstandings and “lack of consensus”, but fail to convince about which sound information and critical considerations about FLR is going to be presented in the following pages.

The rest of the article, not surprisingly, follow the agenda of “forest landscape restoration”, minimizing criticism and presenting a wordy and biased description of how the “FLR” have been implemented around the World. There is no sound discussion based in science and few scientific publications being brought to truly question what “FLR” really is and what is its scientific basis. For instance, one of the main questions about FLR is the risk of threatening non-forest ecosystems with afforestation; in a 72-page document in only three instances grasslands are mentioned, and no discussion is done about it at all.

Given that, I think that authors could step back from the “FLR” agenda and provide a sound scientific review of what have been done under the umbrella of “FLR” and how science can inform the practice of FLR in order to advance with its objectives without harming other environmental and social goals.

===PREPARING YOUR MANUSCRIPT===

===PREPARING YOUR REVISION IN SCHOLARONE===

To revise your manuscript, log into <https://mc.manuscriptcentral.com/rsos> and enter your Author Centre - this may be accessed by clicking on "Author" in the dark toolbar at the top of the

page (just below the journal name). You will find your manuscript listed under "Manuscripts with Decisions". Under "Actions", click on "Create a Revision".

Author's Response to Decision Letter for (RSOS-201218.R0)

See Appendix A.

Decision letter (RSOS-201218.R1)

Dear Dr Stanturf,

It is a pleasure to accept your manuscript entitled "Forest Landscape Restoration: State of Play" in its current form for publication in Royal Society Open Science.

on behalf of Dr Agnieszka Latawiec (Associate Editor)
openscience@royalsociety.org

Appendix A

Response to Reviewers

Reviewer	Comment	Response
R1	The authors compile information, largely based on Bonn Challenge initiatives and organizational information sources, on the state of the art of Forest Landscape Restoration around the world. This task is difficult as there is no clarity on how to define restoration activity as conforming to FLR beyond the labels used to describe it. Furthermore, it is possible for FLR to evolve from more focused projects that take on broader landscape-scape aspects over time; not all projects are planned as fully complete FLR implementation at once. In fact most do not take form that way. So the state of the art is not really possible to assess at a global scale, in terms of FLR"ness." And there are no clear criteria that are useful beyond knowing details of the specific interventions undertaken or planned and how stakeholders are engaged, as the number of trees planted or the number of hectares restored do not uniquely represent FLR. These considerations are important to interpret the many tables included in the paper and compilations of data that are pulled together. What does all of this mean? I'm afraid that we still have a very poor understanding of where and how FLR is being implemented around the world, even with all of this information. So the title is a bit misleading. And the situation will not get any better without more clarity on how to define what is FLR on the ground. This means getting to specific criteria and indicators based on the principles of FLR.	1. We agree with the reviewer that there is some laxity in general in the use of the term FLR and what it includes or not. However, all definitions (and there are several) of FLR as well as principles of FLR, point to some specificities of FLR that distinguish it from other forms of forest/landscape transformation. We try to further explain this in the introduction, the conclusion and in other parts where appropriate. 2. We have changed the title to State of Play as we agree that State of the Art might be misleading.
R1	Page 3, line 21: Major restoration activity, is being conducted in the Philippines, which is not even mentioned here. This is not under Bonn Challenge, but their own initiatives, called the Philippines National Regreening Program. (more info provided in later comment).	We focus on Bonn Challenge countries. The Philippines is not a Bonn Challenge country but we have added some references to the NRP (that has in most cases been a disappointment).
R1	Page 4, line 27: how does an intervention get the FLR label? What does this mean in practice?	Here we are referring to such practices as planting trees in native grasslands, or converting

		native forests to exotic plantations.
R1	Page 4, Line 50: and how to ensure that FLR covers at least three bases (and more) if there are no standards, criteria, or indicators? See Chazdon et al. (2020) for a discussion of principle, criteria, and indicator frameworks for FLR.	There are definition, principles (and recent guiding elements in the ITTO guidelines that provide a lot of guidance on what is in and what is out).
	Page 5, Line 37: this history is also described here: Laestadius, L., K. Buckingham, S. Maginnis, and C. Saint-Laurent. 2015. Back to Bonn and beyond: A history of forest landscape restoration and an outlook for the future. Unasylva 245 66:11-18. also cite: Sabogal, C., C. Besacier, and D. McGuire. 2015. Forest and landscape restoration: concepts, approaches and challenges for implementation. Unasylva 245 66:3-10.	And there are many other references that could be cited here. We feel that we have cited the most relevant.
R1	Page 6, line 10: The GPFLR also made a decision to use "Forest AND Landscape Restoration" as a deliberate step to include non-forest ecosystems in landscape restoration process. This is ignored in this review. It is not a trivial point. FLR is not just about restoring forests at the landscape scale and should not be conflated with landscape-scale reforestation or afforestation or ecological restoration of forests.	The Bonn Challenge uses "forest landscape restoration". No formal decision was ever taken in the GPFLR – some members simply drifted to use one, while others stuck with the original terminology. Most of the GPFLR partners use both (with a predominance of forest landscape restoration).
R1	Page 6, line 10: but there is no system in place for identifying criteria and indicators of these principles to really assess whether FLR is being conducted. See Chazdon et al (2020).	We argue that once you get to the level of criteria and indicators it is going to be location specific. There is no single FLR applicable around the globe. The definition and broad principles are sufficient to determine whether it is FLR. Then whether the practice lives up to expectations is indeed a question of monitoring and indicators will be location specific.

R1	Page 6, line 31-41: This is a very top-down, technocratic vision of FLR. What about the stakeholders who live in the landscape? Who are these experts? They may not really know the landscape or its history and human and natural resources available. The landscape context is critical for FLR planning and implementation. FLR can be based on an existing project or land uses rather than constructed de novo. It can evolve over time and doesn't have to be rolled out all at once. This is a very project and time-scale dependent approach that is unrealistic and does not address long-term goals or progressive changes over time.	We raise these questions lower down in the paragraph. All FLR is ultimately going to be project dependent. Even government initiatives are at the end of the day implemented via projects. Adaptive management is indeed central to the FLR approach, is one of the principles of FLR and is referred to several times in the article.
R1	Page 9, lines 33-39: The ROAM process requires extensive stakeholder engagement at national or subnational level. It is not a conceptual exercise. It is about forming coalitions and getting buy in from multiple sectors. But it is just the beginning and does not get to the implementation stage. It was designed for national level but is now being used to stimulate subnational assessments and implementation.	We state “This methodology is aimed at defining and prioritising opportunities and the course of action for FLR within a national or sub-national context”
R1	Page 9, heading for section 5. Forest restoration is not the same thing as Forest and Landscape Restoration.	This is purposeful, as many of the initiatives that are mentioned are what governments may report under the Bonn Challenge but do not necessarily qualify as FLR as per the definition and principles (see last section where this is discussed).
R1	Page 10, line 50: To be clear, these are not hectares of forest. They are ha of dryland regions where tree cover has increased including agroforestry parklands and enhancing crop production based on FMNR.	We do not claim that these are forests. Additional text has been added to further clarify that we are talking about adding trees to the landscape.
R1	Page 11, lines 6-8: This is why it is forest and landscape restoration. Includes areas that are predominantly	We disagree that a mosaic landscape includes areas that are predominantly grassland. A

	grassland ecosystems.	mosaic landscape in FLR is one where forests are included with other land uses. Grassland may be included but also small communities, farmland, etc. Text has been added to clarify.
R1	Page 12, line 42: Atlantic Forest. the latest assessment using high-resolution imagery shows 28% forest cover (Rezende et al. 2018). Much of the forest here is in small fragments.	This is true but not very relevant here.
R1	Page 12, with regard to Brazil, should also mention that restoration is mandatory, with specific criteria depending on the region and farm area (Chaves et al. 2015, Brancalion et al. 2016)	Statement and reference added. It is also true that this was not rigorously enforced in the past.
R1	Page 13, line 40: should mention here the Philippines National Regreening Program, not a part of the Bonn Challenge, but a significant commitment that has been renewed. (Gregorio and Herbohn 2018, Gregorio et al. 2020, von Kleist et al. 2020). Also FAO has extensive involvement in FLR in the Philippines: http://www.fao.org/3/a-i5412e.pdf#page=135	This is not a Bonn Challenge country. As stated above, we have included references to the Philippines.
R1	Page 22, line 52: please check this figure. I think two digits are missing here?	Thank you, we have corrected it.
R1	Page 25, line 18: It is unclear how these efforts are really based on FLR approach. This paper is conflating rather than clarifying these interventions.	By and large, these efforts predated FLR (in one case by a century). They are included to show that landscape-scale interventions have been around for a long time and in some cases, such as South Korea, are indeed highly regarded as proto-FLR. There is also the mistaken aversion to “afforestation” as a practice because it is mistakenly taken to describe only those misguided efforts where native grasslands were planted to trees. We try to make clear in multiple locations that we do not approve of that either. In any event, we have added some

		text to further describe our intent here.
R1	Page 25, section on Successes and failures: relevant paper by the first author that is not cited and presents similar conclusions (Höhl et al. 2020). Literature Cited: Brancalion, P. H., L. C. Garcia, R. Loyola, R. R. Rodrigues, V. D. Pillar, and T. M. Lewinsohn. 2016. A critical analysis of the Native Vegetation Protection Law of Brazil (2012): updates and ongoing initiatives. Natureza & Conservação 14:1-15. Chaves, R. B., G. Durigan, P. H. Brancalion, and J. Aronson. 2015. On the need of legal frameworks for assessing restoration projects success: new perspectives from São Paulo state (Brazil). Restoration Ecology 23:754-759. Chazdon, R. L., V. Gutierrez, P. H. Brancalion, L. Laestadius, and M. R. Guariguata. 2020. Co-Creating Conceptual and Working Frameworks for Implementing Forest and Landscape Restoration Based on Core Principles. Forests 11:706. Gregorio, N., J. Herbohn, R. Tripoli, and A. Pasa. 2020. A local initiative to achieve global forest and landscape restoration challenge—Lessons learned from a community-based forest restoration project in Biliran Province, Philippines. Forests 11:475 https://doi.org/410.3390/f11040475. Gregorio, N., and J. Herbohn. 2018. Implementing the National Greening Program in the Philippines: lessons learned. Current Conservation 12:25-28. Höhl, M., V. Ahimbisibwe, J. A. Stanturf, P. Elsasser, M. Kleine, and A. Bolte. 2020. Forest Landscape Restoration—What Generates Failure and Success? Forests 11:938. Laestadius, L., K. Buckingham, S. Maginnis, and C. Saint-Laurent. 2015. Back to Bonn and beyond: A history of forest landscape restoration and an outlook	This paper was not accepted at the time we wrote the present paper; we have added it to the revised paper. Agree that some of the conclusions are similar, and they are similar to other of our publications, but we have gone further in the present paper.

	for the future. Unasyuva 245 66:11-18. Rezende, C., F. Scarano, E. Assad, C. Joly, J. Metzger, B. Strassburg, M. Tabarelli, G. Fonseca, and R. Mittermeier. 2018. From hotspot to hopespot: An opportunity for the Brazilian Atlantic Forest. Perspectives in ecology and conservation 16:208-214. Sabogal, C., C. Besacier, and D. McGuire. 2015. Forest and landscape restoration: concepts, approaches and challenges for implementation. Unasyuva 245 66:3-10. von Kleist, K., J. Herbohn, J. Baynes, and N. Gregorio. 2020. How improved governance can help achieve the biodiversity conservation goals of the Philippine National Greening Program. Land Use Policy:104312.	
R2	The article provides an extensive assessment of "Forest Landscape Restoration" ("FLR), but with a low level of novelty and bringing few, if any, critical science issues on the matter. I think that a formal literature review and/or metadata analysis goal should be clearly stated and an objective method of gathering information exposed. The "state of the art" in a science journal should be built on sound science.	We believe that the novelty lies in the collection of data concerning existing initiatives that are being called FLR within Bonn Challenge countries, as well as pointing out where other landscape-scale efforts are underway or have been done in the past. As far as we know this has not been done before. We understand that this reviewer is critical of the FLR approach (and indeed so have we been in other articles, e.g. Mansourian et al. 2017 in the journal Restoration Ecology). However, the purpose of this article is not to assess the value of interventions, but rather to demonstrate their extent and how that compares to the commitments made. For a formal literature review of forest restoration practices, see the Tamm Review by Stanturf, Palik and Dumroese (2014)

		Forest Ecology and Management.
R2	In the summary, authors loosely state that tree planting is “inexpensive” for mitigating climate change, what is questionable, both because it is frequently expensive, and because it has limited contribution in mitigation of present World emissions. Moreover, while the article deals with forests and “forest landscape restoration”, it should not ignore that planting forests will be welcome to reverse landscape degradation and biodiversity loss in former forested areas only, otherwise it will be a form of degradation. Maybe a bit more of context will help, including in this summary what is specifically called as “forest landscape restoration” and acknowledge its limitations in terms of both real contribution for changing climate reality and in which landscapes it is applicable.	We think that the reviewer mistakenly assumes that we think it is an inexpensive way. We clearly state “has been widely touted” that does not mean by us..! As this is the summary, we are not providing references, but there is a multitude of references talking about this.
R2	Introduction apologetically praises planting trees and the FLR” without any critical considerations. The text is built to justify the need of the article being introduced relying in reputed misunderstandings and “lack of consensus”, but fail to convince about which sound information and critical considerations about FLR is going to be presented in the following pages.	Our starting point is that there is a movement to plant trees – that is a fact. We then seek to explore what FLR was set up to achieve (and we have now strengthened that section in the introduction, methods and background) and what is actually happening in practice. We cannot assess the value of what is happening in large part because there is limited monitoring, but also because that is beyond the scope of this work, which seeks to merely compare commitments and reality.
R2	The rest of the article, not surprisingly, follow the agenda of “forest landscape restoration”, minimizing criticism and presenting a wordy and biased description of how the “FLR” have been implemented around the World. There is no sound discussion based in science and few scientific publications being	We are not seeking to judge the value of FLR initiatives, merely to illustrate where they are happening. FLR takes place (as it was defined) in forested landscapes.

brought to truly question what “FLR” really is and what is its scientific basis. For instance, one of the main questions about FLR is the risk of threatening non-forest ecosystems with afforestation; in a 72-page document in only three instances grasslands are mentioned, and no discussion is done about it at all.

There is not scope for FLR in grasslands or other non-forest ecosystems.

We would also like to highlight some sentences in our article that demonstrate that we are not “following an agenda” as implied by this reviewer, but rather note the criticism as well, e.g. in the following:

“The importance of local context adds complexity and nuance that can be overlooked in national and international, top-down restoration programmes”

“although the accuracy of these estimates has been challenged (e.g., [67-70] and the strategy of tree planting for restoration or to mitigate climate change has drawn opposition, particularly in developing countries”

“Nevertheless, these gains would come at the expense of crop and pastureland at a time when there are increasing demands for agriculture and bioenergy [222-224]. Global commitments disconnected from local contexts are a recipe for disappointment”

“In practice, the ‘easiest’ indicator which is predominantly used remains the numbers of trees planted. However, such an indicator does

		not demonstrate persistence of the trees or any actual benefit (social or ecological) from those trees” “This has been apparent in the top down BC commitment process where land area targets are made apparently overlooking local constraints” “Designing FLR as the right trees in the right place avoids some of the criticisms that have included mistaking lack of forest cover as degradation (e.g., where native grasslands occur”
R2	Given that, I think that authors could step back from the “FLR” agenda and provide a sound scientific review of what have been done under the umbrella of “FLR” and how science can inform the practice of FLR in order to advance with its objectives without harming other environmental and social goals.	The “sound scientific review” that this reviewer requests will have to wait until there is more information of what has actually been done under the umbrella of FLR. As we point out, these data are lacking and to our knowledge, there is no effort globally to critically review FLR projects. In some part this is because there are no criteria for evaluating what practices qualify as FLR. Even the few attempts at defining criteria and indicators are to our minds insufficient. NEPCon has proposed a guide for auditors to use to evaluate ecosystem restoration so maybe this is a first start, if there is ever an appetite to “certify” FLR projects. Because the Bonn Challenge is voluntary, lacking

		treaty obligations, such rigor has been opposed.
--	--	--